# Competing basal ganglia pathways determine the difference between stopping and deciding not to go

Kyle Dunovan[1,2†], Brighid Lynch[2,3], Tara Molesworth[2,3], Timothy Verstynen[2,3*†]

[1]Department of Psychology, University of Pittsburgh, Pittsburgh, United States; [2]Center for the Neural Basis of Cognition, University of Pittsburgh and Carnegie Mellon University, Pittsburgh, United States; [3]Department of Psychology, Carnegie Mellon University, Pittsburgh, United States

**Abstract** The architecture of corticobasal ganglia pathways allows for many routes to inhibit a planned action: the hyperdirect pathway performs fast action cancellation and the indirect pathway competitively constrains execution signals from the direct pathway. We present a novel model, principled off of basal ganglia circuitry, that differentiates control dynamics of reactive stopping from intrinsic no-go decisions. Using a nested diffusion model, we show how reactive braking depends on the state of an execution process. In contrast, no-go decisions are best captured by a failure of the execution process to reach the decision threshold due to increasing constraints on the drift rate. This model accounts for both behavioral and functional MRI (fMRI) responses during inhibitory control tasks better than alternative models. The advantage of this framework is that it allows for incorporating the effects of context in reactive and proactive control into a single unifying parameter, while distinguishing action cancellation from no-go decisions.

**\*For correspondence:** timothyv@andrew.cmu.edu

[†]These authors contributed equally to this work

**Competing interests:** The authors declare that no competing interests exist.

## Introduction

When at bat, a baseball player can choose not to swing at the incoming pitch in one of two ways: he can cancel the swing reactively based on external cues, for example, of the estimated position of the ball based on sensory signals, or he can cancel it proactively using internal expectations, for example, a strategy of never swinging at the first pitch (*Aron, 2011*; *Botvinick et al., 2001*; *Braver et al., 2001*). These are generally considered to be separable control signals, with proactive suppression being a slower, action-specific process (*Aron et al., 2011*) and reactive inhibition being a fast global suppression mechanism (*Aron and Poldrack, 2006*; *Aron et al., 2014*). However, it has been argued that a purely reactive form of stopping is unlikely to generalize to many real-world scenarios and, in some cases, response inhibition may result from a combination of proactive and reactive efforts (*Aron, 2011*). Indeed, recent evidence suggests that these processes interact, with proactive signals modulating the efficacy of reactive stopping (*Aron, 2011*; *Chen et al., 2010*; *Forstmann et al., 2008*; *Irlbacher et al., 2014*; *Jahfari et al., 2012*; *Stuphorn and Emeric, 2012*; *van Belle et al., 2014*; *van Maanen et al., 2015*; *Zandbelt et al., 2013*; *Zandbelt and Vink, 2010*). Yet the nature of this interaction remains largely unclear.

An interaction between proactive and reactive decisions makes sense given the shared circuitry associated with both forms of control (*Figure 1*). Proactive signals have been most commonly associated with activity in rostral prefrontal and striatal areas (*Cai et al., 2011*; *Majid et al., 2013*; *van Belle et al., 2014*), suggesting a possible competition of facilitation and suppression signals relegated through the direct and indirect pathways, respectively (*Wei et al., 2015*). In contrast, reactive braking has been linked to premotor and prefrontal areas in the right hemisphere, as well as the

**eLife digest** Imagine you are playing baseball. You can decide not to swing the bat at the incoming ball if you see that it is a wild pitch that will be way outside the strike zone; this is known as reactive control. Alternatively, you may decide not to move because you were coached never to swing at the first pitch (proactive control). It is thought that the brain processes these signals separately with reactive control being a quick way to put the brakes on a planned movement and proactive control being a more specific suppression of unwanted actions. However, some researchers have argued that real-life "inhibitory" control decisions are more likely to be made using a combination of both reactive and proactive signals.

In primates, reactive and proactive signals are both processed by a region of the brain called the basal ganglia. However, it is not clear whether these signals pass through the same set of nerve cells, or whether they use separate sets of cells that run in parallel. Dunovan et al. studied how these signals are processed in the human basal ganglia using a combination of experiments and computational models.

The model assumes that reactive and proactive signals are carried by two pathways that are initially separate but eventually converge in the basal ganglia. If these pathways converge, then proactive control signals should complement reactive decisions to "apply the brakes". For example, if reactive signals suggest that the ball may not come over the home plate, the fact that it is also the first pitch would make it easier for the brain to decide not to swing the bat.

Dunovan et al. tested this model by asking human volunteers to complete two tasks where the decision to respond to a stimulus is made proactively using prior knowledge, or reactively using an explicit stop cue. The experiments also used a technique called functional magnetic resonance imaging (fMRI) to measure the activity in the basal ganglia of each volunteer. Simulations from the model were able to predict the observed patterns of behavior and brain activity in several regions that are key to inhibitory control, including the output of the basal ganglia.

These findings provide a possible mechanism for how reactive and proactive control may interact in the brain. Because of the limitations in imaging the human brain, the next step will be to test whether the model is able to predict the behavior of individual nerve cells in the brains of other animals.

subthalamic nucleus, that together compose the hyperdirect pathway (*Aron et al., 2014*; *Cavanagh et al., 2011b*; *Rodriguez et al., 2006*; *Swann et al., 2012*). While in primates the control signals of the direct, indirect, and hyperdirect pathways all project to the internal segment of the globus pallidus (GPi), the primary output of the basal ganglia, it remains unclear whether these signals converge on the same pool of neurons or run in parallel of each other. At the cellular level, successful reactive inhibition depends on the relative timing of descending execution signals from the striatum and fast-braking signals from the hyperdirect pathway at the output of the basal ganglia (*Schmidt et al., 2013*), suggesting that these two forms of control may, in fact, converge before they continue on to the thalamus.

Here we use knowledge of the circuit-level organization of corticobasal ganglia networks to elucidate the dynamic interactions between proactive and reactive inhibitory control. Using both behavioral measures and functional MRI (fMRI) to confirm predictions from a novel computational model, we evaluate two hypotheses: 1) reactive stopping depends on the state of the execution signal (i.e., the two processes are not independent); 2) proactive no-go decisions occur by modulating the execution process so that it fails to reach its decision boundary, rather than by active cancellation of the decision itself.

## Results

### Competitive dynamics of inhibitory control

The architecture of corticobasal ganglia pathways (*Figure 1A*), along with recent electrophysiological (*Schmidt et al., 2013*) and neuroimaging (*Jahfari et al., 2011*; *Jahfari et al., 2012*;

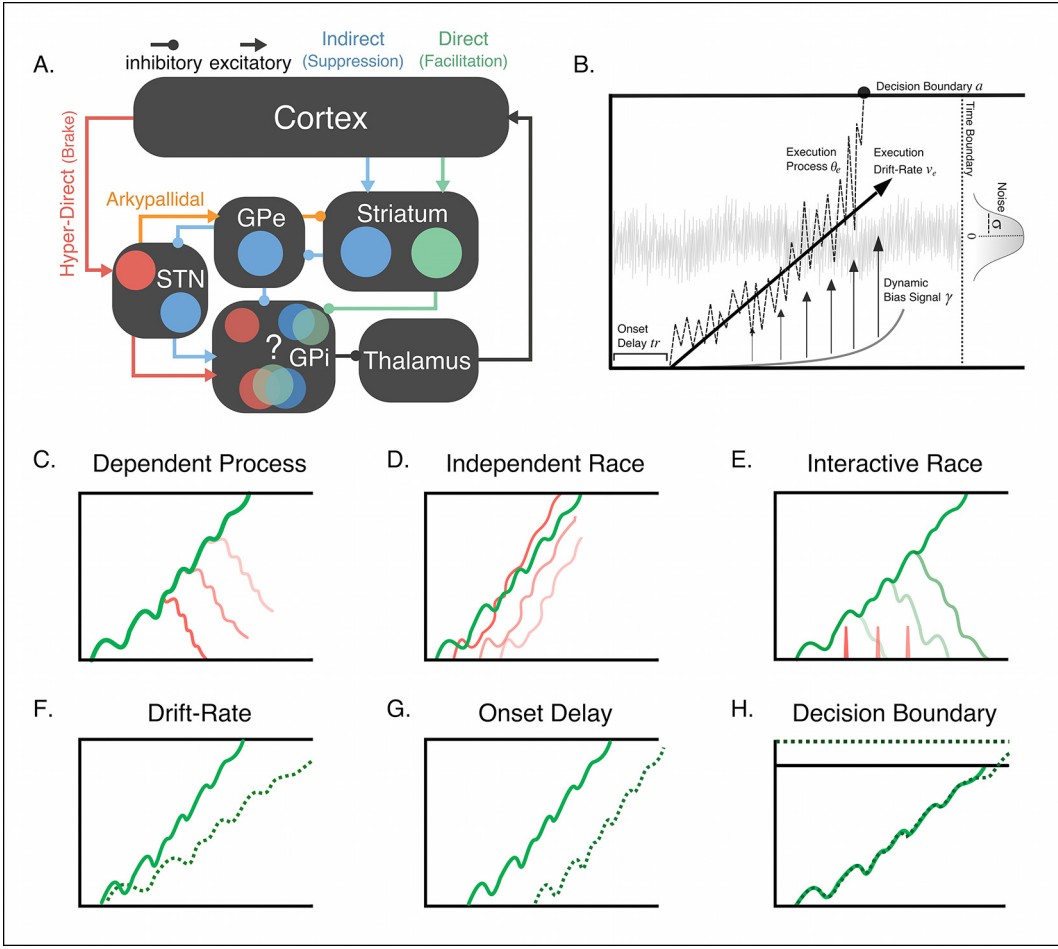

**Figure 1.** Conceptual framework of stopping and deciding not to go as separate but dependent processes. (**A**) The organization of corticobasal ganglia pathways. Execution signals are relayed via the direct pathway (green connections) that result in a disinhibition of thalamic signals to cortex form the internal segment of the globus pallidus (GPi). Thalamic output can be suppressed by one of two major pathways. The indirect pathway (blue connections) increases inhibition of the thalamus via cortical signals terminating on striatal nuclei and relegated through the external segment of the globus pallidus (GPe). The hyperdirect pathway (red connections) quickly increase thalamic suppression via direct projections from the cortex to the subthalamic nucleus (STN). The question mark highlights the uncertainty as to whether the hyperdirect signals terminate on the same GPi neurons as the direct and indirect pathways. (**B**) Parameter structure of the general drift-diffusion process used in all model simulations and fits. For the reactive task (see *Figure 1*), we compared three models, a dependent process model (**C**), a traditional independent race model (**D**), and interactive race model (**E**). See main text for details on these models. For both reactive and proactive tasks, we compared three possible influences of context on the execution process: drift rate modulation (**F**), boundary shift modulation (**G**), and onset delay modulation (**H**). See main text for descriptions of each model.

*Jahfari et al., 2010*; *Smittenaar et al., 2013*; *Zandbelt et al., 2013*) evidence, suggests that the efficiency of fast hyperdirect braking depends on the state of the descending execution process that originates in striatal pathways. To capture this, we designed a novel decision model with nested drifting diffusion processes (*Figure 1B*, see "Materials and methods" section). The motor decision is modeled as a single variable (hereafter, referred to as the execution process) that is assumed to reflect the differential activity between the direct (facilitation) and indirect (suppression) pathways; that is, as facilitatory systems are recruited more than the suppressive systems, the execution process accumulates toward the decision boundary. The cancellation decision (hereafter, referred to as the braking process) is modeled as a latent competing signal, assumed to originate from the hyperdirect (brake) pathway and whose starting state depends on the state of the facilitation

versus suppression competition (*Figure 1C*). In this dependent process model (DPM), a response is realized only if the execution process reaches its decision boundary before the braking process reaches its decision boundary and before the trial time window has expired. Otherwise the model does not produce a response. Also, since it has been shown that the physiological dynamics of inhibitory control networks are best reflected by models where the race to the decision boundary has dynamic, nonlinear properties (*Ratcliff and Frank, 2012*), our model also includes a dynamic gain that dynamically accelerates the execution process as it approaches the time boundary (i.e., the deadline for making a response).

We should note that others have proposed similar nested process models for inhibitory control decisions. Indeed, our DPM is conceptually similar to the interactive race model (*Boucher et al., 2007*), whereby a response is suppressed by the onset of a braking signal that directly inhibits the execution process, preventing it from reaching the action threshold. The interactive race and DPMs make fundamentally different assumptions about the nature of the interaction between the execution and braking processes. The predictions of both the dependent process and interactive race models can be contrasted against the traditional independent race model (*Logan et al., 1984*; *Logan et al., 2014*) where the outcome of the decision is determined by a parallel race between the execution and braking processes.

To evaluate the different reactive stopping models we tested a group of subjects (N = 60, see Materials and methods section) on a modified version of the stop-signal task that allows for precise timing of the target response time (RT) (*Coxon et al., 2012*; *Lappin and Eriksen, 1966*; *Zandbelt et al., 2013*). The task parameters were specifically designed to allow for a competition between proactive and reactive decisions, rather than adapted on a trial-by-trial basis in order to isolate only reactive stopping decisions. On each trial, subjects saw a blue bar 'fill' upwards toward a target line on the screen at a constant rate (*Figure 2A*, see 'Materials and methods' section). On go trials, the bar would intersect the target line at 500 ms after trial onset. Participants were instructed to stop the bar by pressing a key. The closer the bar was stopped to the target line, the higher the financial reward on that trial. In a subset of trials (stop trials; 45% trials), the bar would stop before it intersected with the target (i.e., the stop signal) and participants were instructed to not make a response on these trials. Successful decisions were rewarded and incorrect decisions were penalized. All participants were run in two versions of the experiment: a baseline condition (baseline) and a condition where the feedback signals were more restrictive on accuracy to the target and higher penalties for failing to stop when required while higher rewards were given for successful stops (caution). Similar manipulations on feedback have been shown to modulate stopping performance in the stop-signal task *Leotti and Wager, 2010*; Shenoy and Yu, 2011). As expected, in both conditions it was harder for participants to withhold their responses when the stop signal was delivered later in the trial (*Figure 2B*). In addition, changing the structure of the feedback promoted more cautious responding, resulting in more successful stopping at later stop signal delays (SSDs) and slower RTs on go trials.

Using both the RT distribution and the stop probability curve in the baseline condition, we fit the experimental data to our DPM, the interactive race model, and the traditional independent race model (see 'Materials and methods' section). The DPM provided a better fit to both the correct and incorrect RT distributions (*Figure 3A*) than either the interactive or independent race models. In addition, the DPM fell within the 95% confidence interval of the mean probability of stopping in each SSD condition (*Figure 3B*), whereas the independent and interactive race models overpredicted the probability of stopping at the longest and shortest delays, respectively. Goodness of fit, assessed using both the Akaike information criterion (AIC) and Bayesian information criterion (BIC), showed that the DPM was a substantially better fit to the data than the other two models (*Figure 3C*; *Table 1*), suggesting that the race between braking and execution signals may be biased depending on the state of the action selection decision.

Contextual factors, such as reward contingencies (*Leotti and Wager, 2010*) and cued expectations of stopping probability (*Jahfari et al., 2012*; *Smittenaar et al., 2013*; *Zandbelt et al., 2013*) can influence stopping performance by modulating the speed of the execution decision. We see this same pattern in our caution manipulation, where the magnitude of feedback provides larger rewards for correct stopping and the window for reward on go trials is narrowed (see "Materials and methods" section). Overall, in the caution condition subjects shifted their stop curve later in time, producing more accurate stopping performance (*Figure 2B*; t (59) = −2.18, p=0.03) by delaying

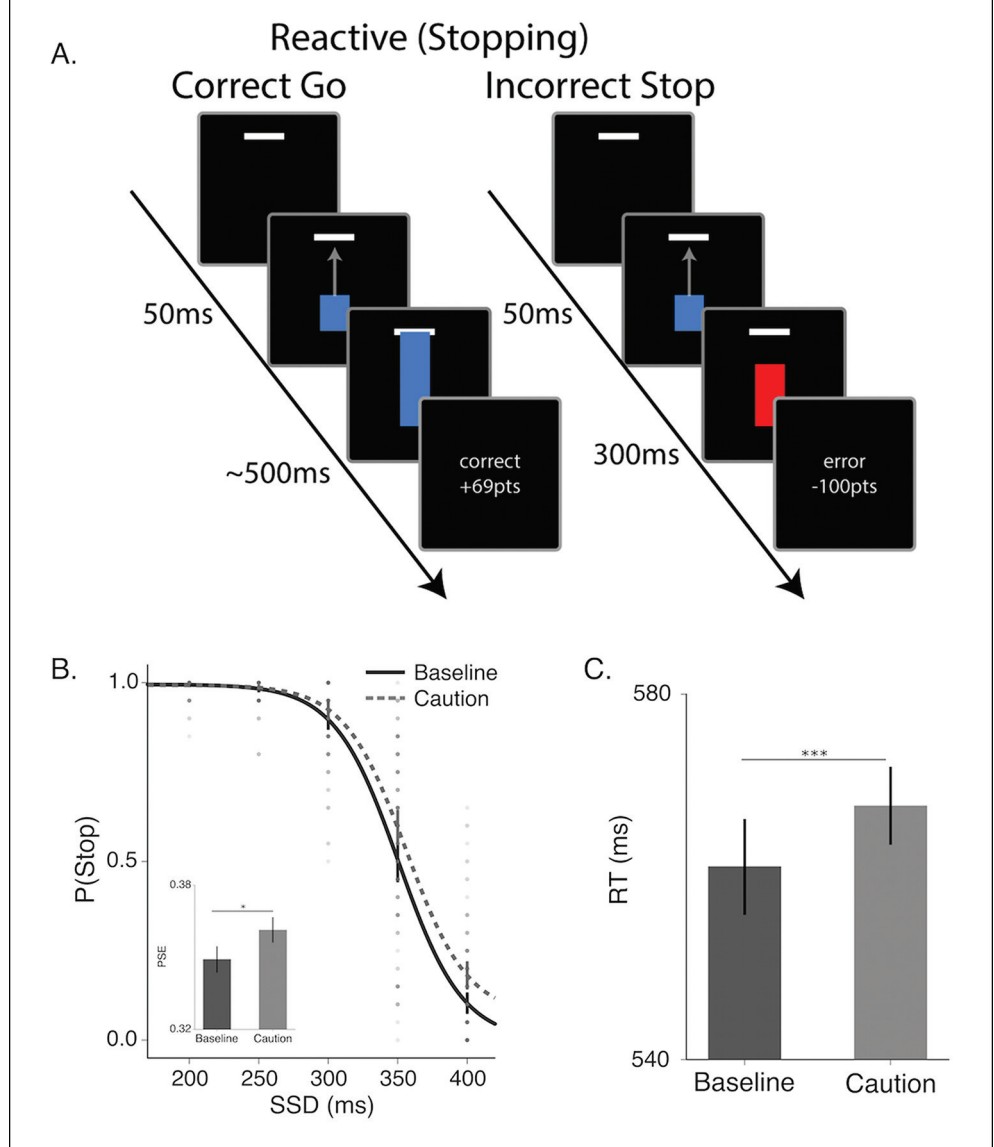

**Figure 2.** Reactive stopping task and behavioral results. (**A**) Timeline of a go (left) and stop trial (right) in the reactive experiment. (**B**) Mean observed probability of stopping in the baseline (dark, solid) and caution (light, dotted) conditions; dots reflect individual subject means. As the stop-signal delay (SSD) increased, the probability of correctly stopping decreased. Inset in *B* shows the mean point of subjective equality (PSE) for both conditions. (**C**) Mean response times (RT) for the baseline and caution conditions. All error bars reflect the 95% confidence interval, * indicates p <0.05, *** indicates p <0.001.

their execution decision, resulting in longer go trial RTs (*Figure 2C* $t(59) = -0.53$, p <0.001). Thus, our feedback manipulation was effective at inducing more conservative decisions.

This contextual effect on the execution process can manifest in several ways. Context can modify the rate of drift (Drift Modulation; *Figure 1F*; *Hanks et al., 2014*; *Standage et al., 2011*; *Zhang and Rowe, 2014*), shift the onset time at which the execution process begins to accumulate (Onset Modulation; *Figure 1G*; *Pouget et al., 2011*), change the distance to the threshold (Boundary Modulation; *Figure 1H*; *Boehm et al., 2014*; *Cavanagh et al., 2011a*; *Forstmann et al., 2012*; *Jahfari et al., 2012*; *Wiecki and Frank, 2012*), or modulate a combination of parameters (*Heitz and Schall, 2012*), such as both drift and onset modulation together. As mentioned above, if direct and indirect pathways converge at the GPi, this would predict that the execution process is driven by an active competition between facilitation and suppression signals, which is naturally implemented in

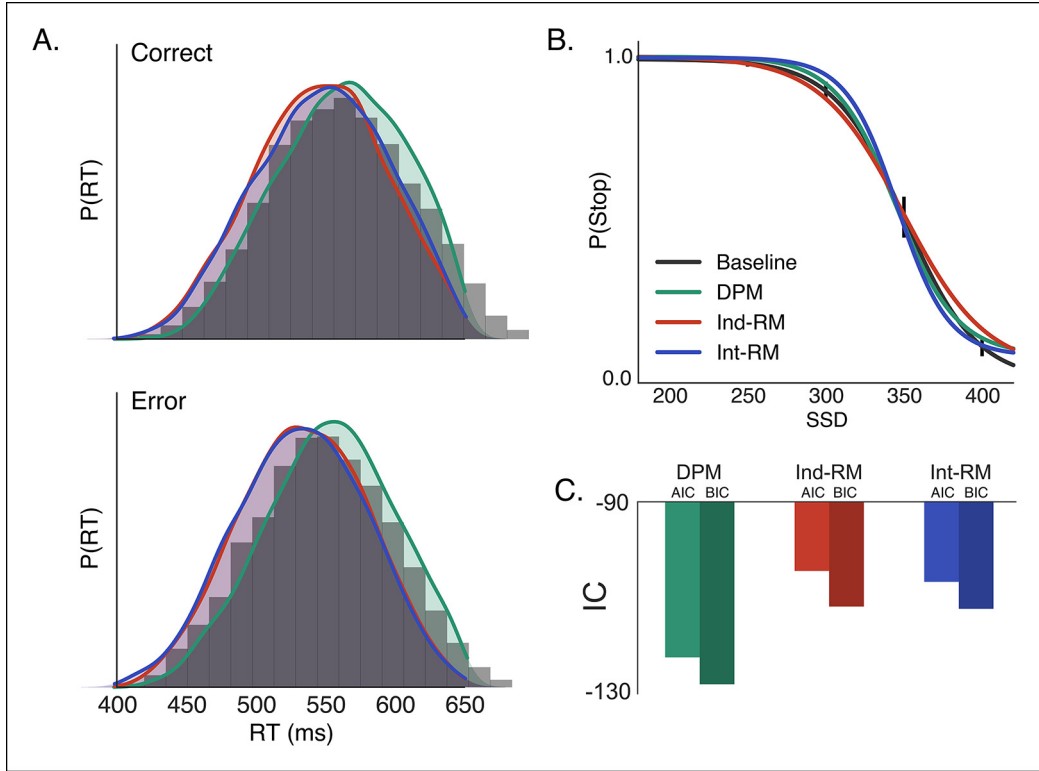

**Figure 3.** Comparison of reactive stopping models. Fits of the three reactive models (*Figure 1C–E*) to behavioral data in the baseline condition, shown against (A) the histogram of RTs for correct (top) and incorrect (i.e., responses made on stop trials; bottom) trials and (B) the stop probability curve. Overall the dependent process model (DPM; green) explains speed and accuracy in the behavioral data (gray histograms in *A*, black line in *B*) better than the independent race model (Ind-RM; red) and interactive race model (Int-RM; blue). (C) Bars show the calculated AIC for each model. Error bars on behavioral data in *B* show the 95% confidence interval of the mean.

diffusion models with the drift rate parameter which captures a moment-by-moment competition between a hypothesis and its alternative.

To evaluate these possibilities, we fit four versions of our DPM in which either the drift, onset, onset and drift, or boundary height parameters were left free when fitting the baseline and caution conditions separately. Qualitatively all models appeared to fit the stop probability curves quite well (*Figure 4—figure supplement 1*). Where the different models varied was in their ability to capture subtle variation in the RT distributions. As a result, the drift modulation model provided the best overall fit of speed and accuracy in the reactive task, followed by the onset modulation and boundary models, respectively (*Figure 4*; *Table 1*). Although the drift modulation model best explained contextual effects on reactive stopping performance, the fact that the onset and boundary modulation models also had very good fits suggests that this manipulation may not cleanly be able to distinguish between the different models. We return to this issue in the next section.

## Not going as a failure to reach bound

The influence of contextual factors on decision dynamics, like demands for control as shown above, as well as evidence that execution uncertainty slows reaction times (*Jahfari et al., 2012*; *Zandbelt et al., 2013*; *Zandbelt and Vink, 2010*), both point to a novel mechanism of no-go decisions that distinguishes them from reactive stops. If the execution signal is determined by a dynamic competition between direct (facilitation) and indirect (suppression) signals (*Figure 1A*; see also [*Wei et al., 2015*]), then making a decision to not execute an action may reflect a scenario where the indirect pathways provide enough active suppression of the direct pathway so that no response is triggered. In other words, a proactive no-go decision occurs when the system fails to accrue enough evidence to reach the execution decision boundary.

**Table 1.** Reactive model parameter estimates and fit statistics. In the top panel, best fit parameter estimates for boundary height (*a*), onset delay (*tr*), execution drift rate (*ve*), braking drift rate (*vb*), dynamic bias gain (*xb*) are listed for each of the candidates reactive stopping models. Additionally, the stop signal onset delay (*sso*) was estimated for the interactive race model but was not included in the other models. The lower panel contains parameter estimates and fit statistics for the candidate models of contextual modulation between baseline and caution conditions of the reactive task. Parameters that were left free to vary between conditions contain two values, one estimate for the baseline condition and another estimate for the caution condition (see *Model fitting* for details regarding acquisition of constant parameters and optimization across conditions). In both panels, the last three columns show the $\chi^2$ as an absolute index of how well each model fit the data as well as the Akaike information criterion (AIC), and Bayesian information criterion (BIC) as complexity penalized goodness-of-fit measures. Lower values in all three measures imply a better fit to the data.

| Model | $a$ | $tr$ | $v_e$ | $v_b$ | $sso$ | $xb$ | $\xi^2$ | AIC | BIC |
|---|---|---|---|---|---|---|---|---|---|
| DPM | 0.534 | 0.174 | 1.266 | −0.990 | | 0.878 | 0.0028 | −122.40 | −128.018 |
| Ind-RM | 0.250 | 0.338 | 1.127 | 1.269 | | 1.52 | 0.0075 | −106.652 | −112.270 |
| Int-RM | 0.445 | 0.220 | 1.195 | 3.023 | 0.197 | 1.474 | 0.0069 | −104.379 | −111.815 |
| Drift | 0.536 | 0.178 | B: 1.289<br>C: 1.243 | −0.984 | | 0.877 | 0.0051 | −273.459 | −273.301 |
| Onset | 0.531 | B:0.171<br>C: 0.180 | 1.236 | −0.960 | | 0.893 | 0.0054 | −271.651 | −271.492 |
| Drift and onset | 0.538 | B: 0.173<br>C: 0.178 | B: 1.269<br>C: 1.247 | −0.989 | | 0.858 | 0.0063 | −260.793 | −263.941 |
| Bound | B: 0.525<br>C: 0.551 | 0.178 | 1.268 | -0.984 | | 0.878 | 0.0057 | −269.994 | −269.835 |

**DPM:** dependent process model; Ind-RM: independent race model; Int-RM: interactive race model.

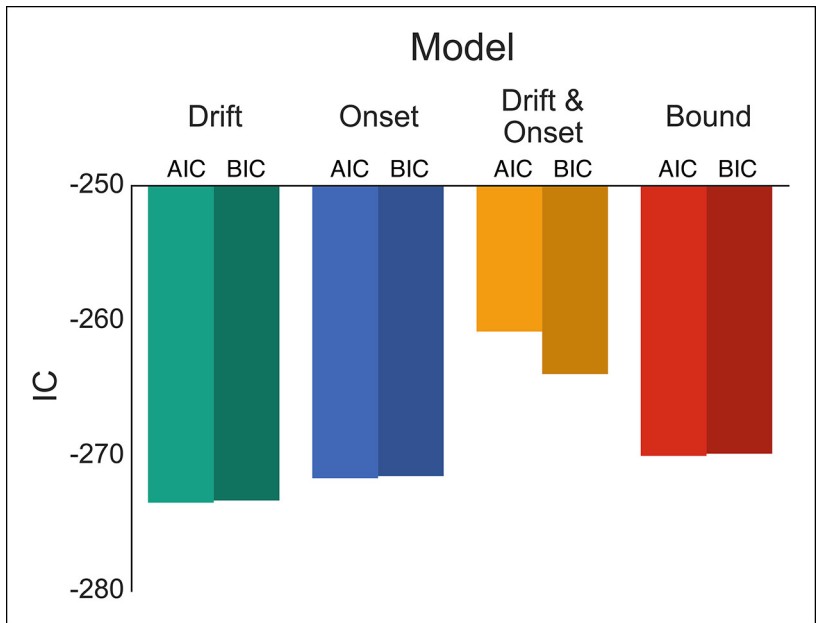

**Figure 4.** Comparison of modulation models in reactive task. Goodness-of-fit measures for execution modulation models in reactive task. Bars show the estimated Akaike information criterion (AIC) and Bayesian information criterion (BIC) for the drift, onset, combined drift and onset, and boundary modulation models. The model with the lowest score, in this case the drift modulation model, is preferred.

The following figure supplements are available for Figure 4:

**Figure supplement 1.** Model predictions of behavior in reactive task.

To evaluate this hypothesis we tested the same group of subjects in a novel proactive inhibition task (see also *Zandbelt et al., 2013*). The overall design was largely similar to the reactive paradigm with the following major changes. First, only one SSD was used. This occurred at 450 ms and is too late for it to be used as a reactive stopping cue. Second, on each trial the color of the rising bar indicated the probability that the stop signal would be delivered (*Figure 5A*): a purely red bar indicates 0% go probability (i.e., stop signal would be presented on every trial), purely blue bar indicates 100% go probability (i.e., stop signal would never be presented), and shades of purple indicate mixture probabilities. Rather than rely on the very late reactive stop signal, subjects were told to use the

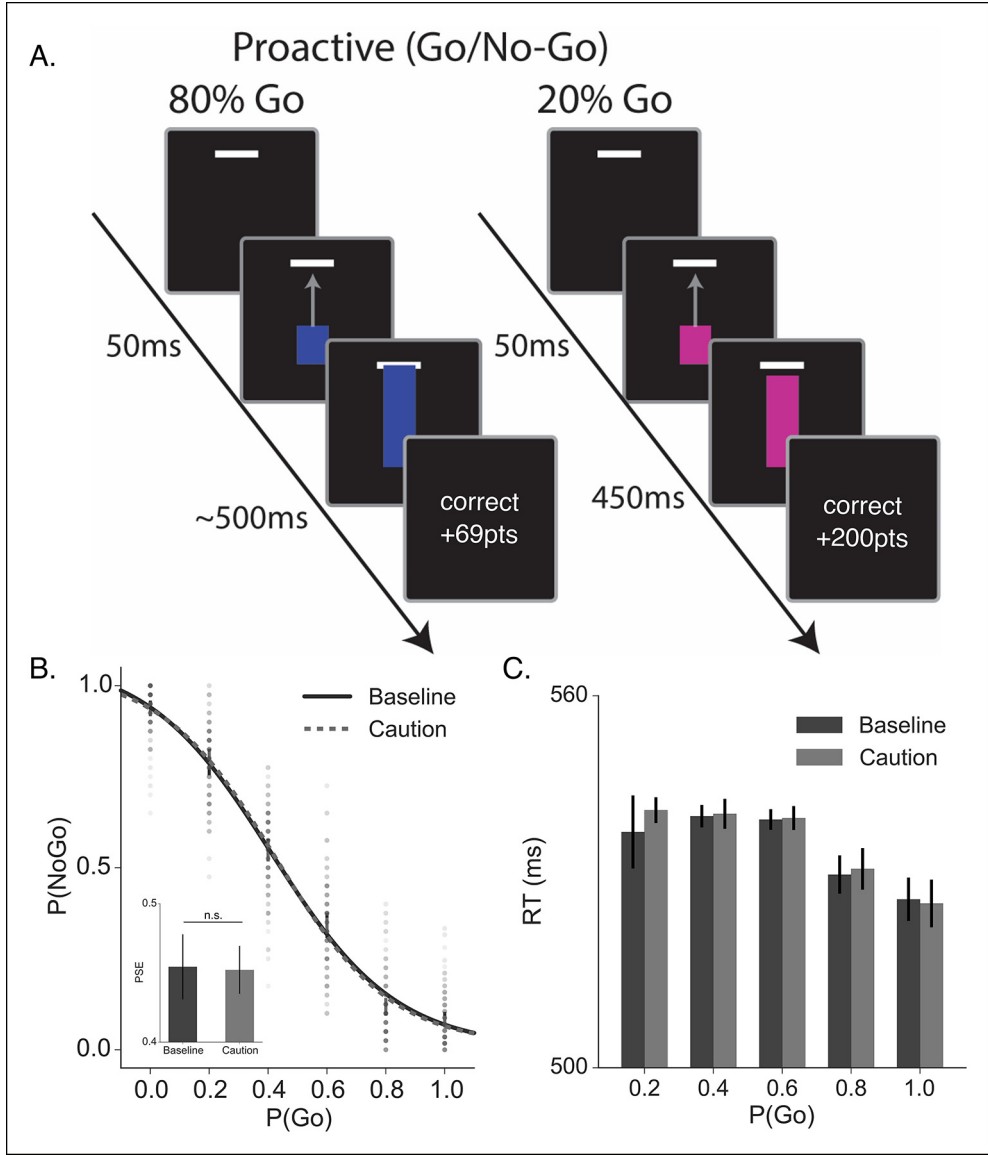

**Figure 5.** Proactive no-go decision task and behavioral results. (A) Timeline of correct high (left) and low (right) go probability trials in the proactive task. The low probability example shows a trial in which a stop signal was presented. (B) Mean probability of making a no-go decision; dots reflect individual subject means and (C) mean reaction time plotted as a function of go trial probability in the baseline (dark, solid) and caution (light, dotted) conditions. Inset in *B* shows the mean point of subjective equality (PSE) for both conditions. All error bars reflect the 95% confidence interval; n.s. indicates a nonsignificant comparison at α of 0.05.

The following figure supplements are available for Figure 5:

**Figure supplement 1.** Subject-wise correlation of reactive and proactive control.

proactive cue to decide whether to execute the response or withhold it. Since subjects were asked to make an informed guess, all our analysis collapsed across both correct and incorrect trials.

Participants were able to use the colored cue to make accurate go and no-go decisions (*Figure 5B*). There was a slight bias toward making a go decision across trial types; however, the probability of making a no-go decision scaled monotonically with the cued probability of a go trial. Also, as expected, participants were faster with higher certainty of a go trial (*Figure 5C*). It is worth noting that all RTs in this proactive task were much faster than the responses in the reactive task (*Figure 2*), confirming that subjects weren't relying on the delayed stop signal to make a reactive decision. Thus, this second experiment was able to get subjects to make a fundamentally different decision than in the reactive task, while keeping most other experimental conditions (e.g., sensory signals) constant. Unlike the reactive task, we did not observe a significant influence of demands for control on either the decision curve ($F[3.43, 205.8] = 0.76$, $p=0.53$) or response times ($F[2.28, 132.32] = 0.81$, $p=0.46$) in the proactive task.

If our hypothesis is correct that reactive stopping ability depends on the current state of proactive control, then there should be a mild degree of shared variance between an individual's reactive stopping ability and their proactive decisions. To test this we correlated the points of subjective equality (PSE) across subjects, between the reactive (*Figure 2B*) and proactive (*Figure 5B*) conditions. The PSE reflects the point at which the psychometric curve crosses 50% chance (see "Materials and methods" section). Collapsing across both the baseline and caution conditions, we found a small but significant correlation in a subject's reactive control abilities and their proactive decisions (*Figure 5—figure supplement 1*, $r = 0.35$, $p < 0.006$, $r^2 = 0.06$). Thus, overall reactive task performance explains ~6% of the variance in the proactive condition, which is consistent with a model where these two abilities reflect separable upstream control signals that eventually converge on common output pathways (*Figure 1A*).

One advantage of our proactive task is that it can better distinguish the mechanisms of modulation on the execution process. Therefore, we fit the same four models that were fit against the reactive task (*Figure 1H*), leaving the specific model parameters free across each go trial probability condition (see "Materials and methods" section). Since there were no differences between the baseline and caution conditions, all data was collapsed together to increase statistical power of the model fits. We found that the drift modulation model was able to capture the proactive no-go decision dynamics significantly better than the other models based on the AIC and BIC scores (*Figure 6*; *Table 2*) and general patterns of speed and response proportions (*Figure 6—figure supplement 1*). This means that as the go trial probability decreased, the drift rate of the execution process decreased to the point that the decision boundary was often not intersected by the end of the trial. Thus, we can reliably capture the dynamics of a no-go decision as a modulation of competing facilitatory and suppressive signals that results in a failure to accumulate enough evidence to make a decision. This provides a fundamental distinction between the control mechanisms of stopping and deciding not to go.

## Accumulating basal ganglia output during no-go decisions

Because the assumptions of our DPM it makes very specific predictions about the physiological dynamics of go versus no-go decisions within corticobasal ganglia networks, specifically in regions where these pathways are known to interact, that is, striatum and thalamus. Previous fMRI studies have reliably shown that the magnitude of activation in cortical decision-making areas approximates the cumulative sum of evidence over time as the decision process approaches its threshold (*Basten et al., 2010*; *Ho et al., 2009*). While the temporal and spatial imprecision of fMRI precludes direct comparison of neural and computational mechanisms (*Simen, 2012*), the overall magnitude of the signal can be taken as a proxy for the duration and distance-to-bound of the decision process (*Forstmann et al., 2010*). By integrating the execution process over time we compared model-predicted activation levels for both go and no-go decisions (see "Materials and methods" section) with the task-evoked blood oxygen level dependent (BOLD) response measured by fMRI in corresponding conditions of our proactive task. Simulations were run for the drift, onset, and boundary modulation models of proactive control (see *Figure 7A–C*).

Based on our simulations, the drift modulation model predicts a greater BOLD response when an action is executed (i.e., 'go') versus trials where the action is withheld (i.e., 'no-go') and, more importantly, that the BOLD response should increase with the go trial probability during 'no-go' decisions,

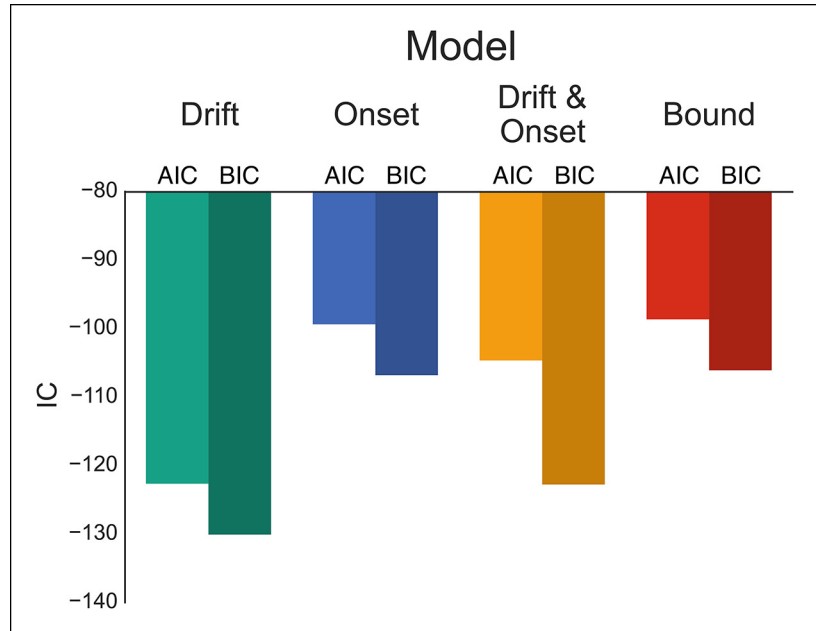

**Figure 6.** Comparison of modulation models in proactive task. Goodness-of-fit measures for execution process modulation models in proactive task. Same plotting conventions as in *Figure 4*. As with the reactive experiment, the drift modulation model provided better fits to the proactive task data than the other models.

The following figure supplements are available for Figure 6:

**Figure supplement 1.** Model predictions of behavior in proactive task.

but taper off when a response is executed. The main effect of trial outcome (i. e., 'go' greater than 'no-go') is due to the fact that, on average, withheld responses result from lower levels of evidence accumulation than executed responses (i.e., trials in which the accumulated evidence reaches the threshold). Within 'no-go' decision trials, the increase in drift-rate with higher go-trial certainty leads to higher levels of sub-threshold evidence accumulation at the end of the trial. On the other hand, the increase in drift-rate with go-trial certainty leads to a faster rise-to-threshold on 'go' decision trials and thus produces slightly less summed activity than executed responses in lower go-trial probability conditions. In contrast with the drift modulation model, the onset modulation model predicts that in addition to a main effect of trial outcome, increasing the probability of a go-trial should increase the magnitude of the BOLD response for both 'no-go' and

**Table 2.** Proactive model parameter estimates and fit statistics. Best fit parameter estimates for boundary height (*a*), onset delay (*tr*), execution drift rate (*$v_e$*), and dynamic bias gain (*xb*) are listed for each of the proactive modulation models. For each model, free parameter (s) are named in the shaded column, followed by fitted estimates in each go trial probability condition in columns $P_0$–$P_{100}$, (see *Model fitting* for details regarding acquisition of constant parameters and optimization across conditions). The last three columns show the $\chi^2$ as an absolute index of how well each model fit the data as well as the Akaike Information Criterion (AIC), and Bayesian Information Criterion (BIC) as complexity penalized goodness-of-fit measures. Lower values in all three measures imply a better fit to the data.

| Model | a | tr | $v_e$ | xb | | $P_0$ | $P_{20}$ | $P_{40}$ | $P_{60}$ | $P_{80}$ | $P_{100}$ | $cX_\xi^2$ | AIC | BIC |
|---|---|---|---|---|---|---|---|---|---|---|---|---|---|---|
| Drift | 0.487 | 0.292 | | 1.563 | $v_e$ | 1.411 | 1.562 | 1.683 | 1.761 | 1.880 | 1.925 | 0.0022 | −122.65 | −130.08 |
| Onset | 0.628 | | 1.42 | 0.641 | tr | 0.182 | 0.161 | 0.134 | 0.117 | 0.084 | 0.076 | 0.0095 | −99.38 | −106.81 |
| Drift and onset | 0.06 | | | 1.468 | $v_e$ | 0.831 | 0.970 | 0.968 | 0.979 | 0.932 | 1.079 | 0.0033 | −104.65 | −122.77 |
| | | | | | tr | 0.515 | 0.506 | 0.492 | 0.479 | 0.451 | 0.463 | | | |
| Bound | | 0.272 | 0.914 | 0.913 | a | 0.379 | 0.344 | 0.305 | 0.281 | 0.246 | 0.236 | 0.0099 | −98.65 | −106.09 |

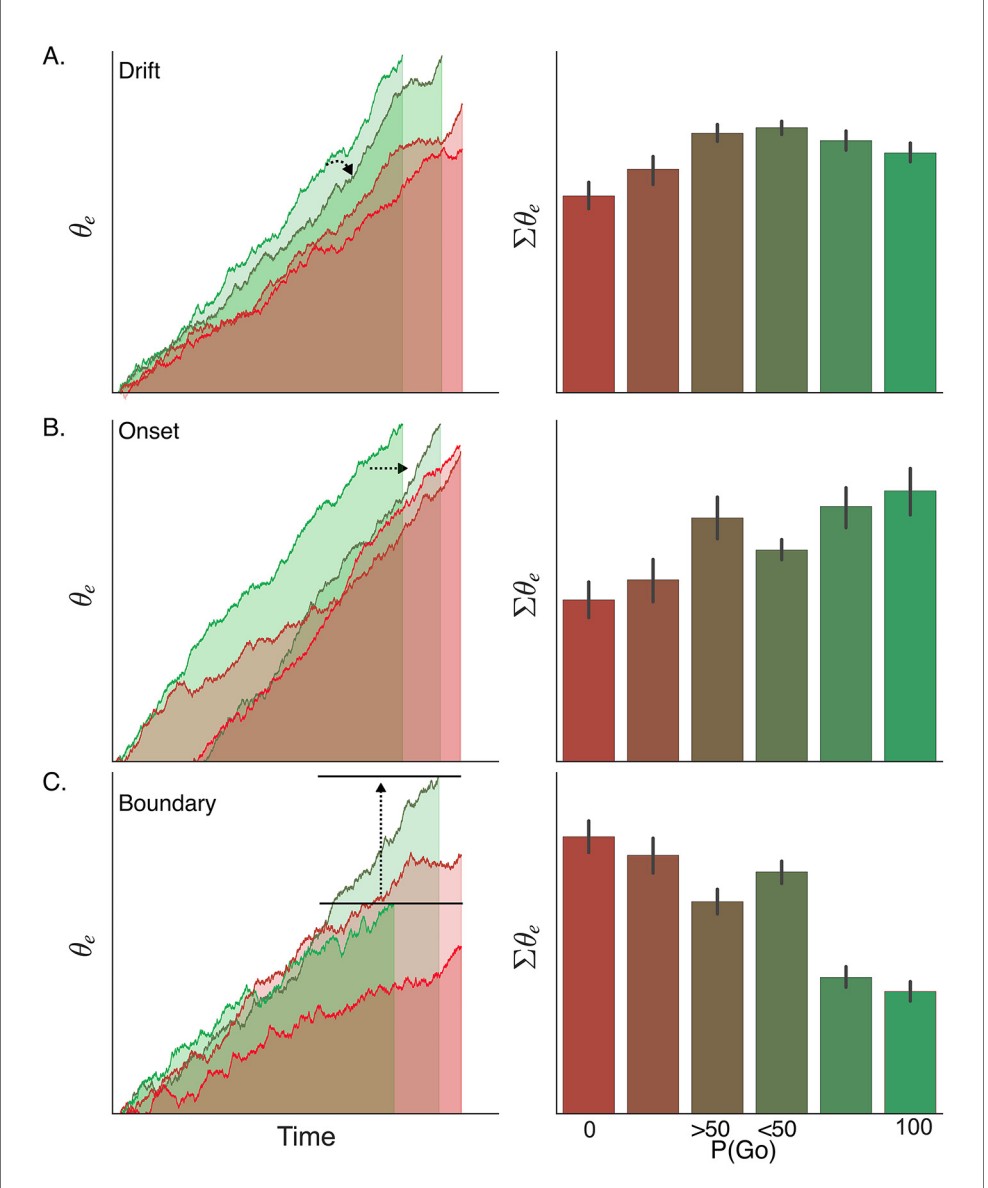

**Figure 7.** Simulated interaction between trial outcome and response expectation on BOLD activation. Time-course of BOLD and mean activity in the proactive task predicted by the (**A**) drift modulation, (**B**) onset modulation, and (**C**) boundary modulation models. Time courses (left column) reflect the rise of the execution process ($\theta_e$) on 'go' trials (left panels; green lines) and 'no-go' trials (right panels; red lines). Lighter colors reflect lower go probability conditions (go, <50%; no-go, 0%) and darker colors represent higher go probability conditions (go, 100%; no-go, >50%). The arrows in *A* and *B* show the pressure on the execution process with decreasing go trial certainty. The arrow and bars in *C* illustrate the shift in boundary height between high (bottom bar) and low (top bar) go trial probability conditions. Mean simulated BOLD responses (right column) are calculated as the cumulative sum of execution process across the full time course. This summation is reflected in the filled area under the curve in the traces in the left column. BOLD: blood oxygenation level dependent.

'go' decision trials. This is due to the fact that there is overall more time for the process to accumulate with earlier onsets, that is, higher go-trial probabilities. Thus, in high go-trial probability conditions both 'go' and 'no-go' decisions arise from longer cumulative processing time than in low go-trial probability conditions, and therefore predict greater activation. This model also predicts the effect of go-trial probability should be roughly equivalent for withheld and executed responses, which is a contrasting prediction from the drift modulation model which predicts a greater

modulation wtih go-trial probability on 'no-go' decision trials. Finally, the boundary modulation model predicts that increasing go-trial probability will decrease the BOLD response, along with a general decrease in BOLD signal during 'go' decisions. This is because conditions with lower boundary heights (i.e., greater certainty of a go-trial), allow less time for the process to be active and a shorter distance to travel and thus predict lower levels of summed activity.

To evaluate these different model predictions, we measured task-related hemodynamic responses using the fMRI BOLD contrast. Healthy subjects (N=20) performed a modified version of the proactive task for the MRI environment (see "Materials and methods" section). In order to assure that no covert actions were elicited during trials in which a key press was not recorded, we used a modified EMG over the first dorsal interosseous (FDI) muscle of the responding hand to record muscle activity (*Figure 8—figure supplement 1A*) so as to identify finger movements on all trials. Viable EMG data was available for 14 subjects and in these subjects suprathreshold muscle activity was only observed in <1% (mean = 0.71%, max 4%) of the trials where no key press was detected. The number of missed key presses was equally distributed across go trial probability conditions. For the initial analysis these error responses were corrected in these subjects using the EMG data to assure that no-go trials were uncontaminated by covert actions in these subjects.

For the purpose of testing the predictions of the drift, boundary and onset modulation models, we used a general linear model (GLM) to fit task-related responses during 'go' and 'no-go' trials separately and a parametric regressor for each condition to assess the effect of go trial probability on the evoked BOLD response (see "Materials and methods" section). Using a random effects analysis on all subjects, we isolated regions whose activity was modulated more during no-go trials than during trials where a response was executed, as predicted by the drift modulation model. After adjusting for cluster size (k > 40) and multiple comparisons (q < 0.05) we identified 24 clusters with differential modulation during no-go responses than go responses (*Supplementary file 1*). We were particularly interested in a cluster in the right caudate nucleus and a pair of bilateral clusters in the ventromedial thalamus that are consistent with the location of the thalamic subnucleus that is reciprocally connected to motor cortex (*Nambu, 2011*).

The nature of the contrast used in the whole-brain analysis does not necessarily confirm that the pattern of responses in the right caudate and bilateral thalamus are consistent with the drift modulation model, as the effect of go trial probability could cause a greater decrease in activation during no-go trials (e.g., the opposite direction of the drift modulation predictions). Therefore, we extracted out the condition-specific responses in each region of interest (ROI), for all subjects, using a fivefold cross-validation with anatomical priors so as to avoid performing circular inference. Consistent with our predictions, activity in the right caudate nucleus and the left and right thalamic ROIs show a main effect of decision type on evoked activity (*Figure 8*; caudate: $F_{(1,32)} = 2.69$, p=0.03; left thalamus: $F_{(1,32)} = 4.01$, p=0.003; right thalamus: $F_{(1,32)} = 4.50$, p=0.001). In the thalamus ROI responses selectively scaled with go trial probability, even in the trials where no key press was detected (*Figure 8*; left: $F_{(2,32)} = 4.13$, p=0.025; right: $F_{(3,32)} = 3.67$, p=0.037), while this effect was not significant in the caudate ROI ($F_{(2, 32)} = 1.77$, p=0.19). More importantly the interaction between go trial probability and decision type was significant for both thalamus ROIs (left: $F_{[2,32]} = 4.30$, p=0.022; right: $F_{[2,32]} = 4.13$, p=0.025), but not the caudate ($F_{[2,32]} = 1.22$, p=0.31). It should be noted that these results do not significantly change when only the subjects with reliable EMG signals are included in the analysis, confirming that spurious movements do not drive this pattern.

Two cortical regions, the inferior frontal gyrus (IFG) and supplementary motor area (SMA) near the preSMA, were also of particular interest because they have been previously implicated in inhibitory control (*Figure 8*). In the IFG, we observed a lateralized interaction between go trial probability and response type in the right hemisphere ($F_{[2,38]} = 4.05$, p=0.027), but not in the left ($F_{[2,38]} = 1.45$, p=0.25). This rightward lateralization is consistent with previous reports of the right IFG being implicated in control of the hyperdirect pathway (*Aron et al., 2014*; *van Belle et al., 2014*; *Zandbelt and Vink, 2010*). The SMA also exhibited an interaction between go trial probability and response type in both the right ($F_{[2,38]} = 4.86$, p=0.014) and left ($F_{[2,38]} = 4.52$, p=0.019) hemispheres. These results are consistent with previous studies implicating SMA and preSMA in uncertainty-related and conflict-related modulation of boundary height in the traditional drift-diffusion model, achieved through connections with the striatum and subthalamic nucleus (STN) (*Forstmann et al., 2010*; *Frank et al., 2015*; *Jahfari et al., 2012*). This finding supports the notion that modulating the execution drift-rate in our model reflects similar computations as adapting the

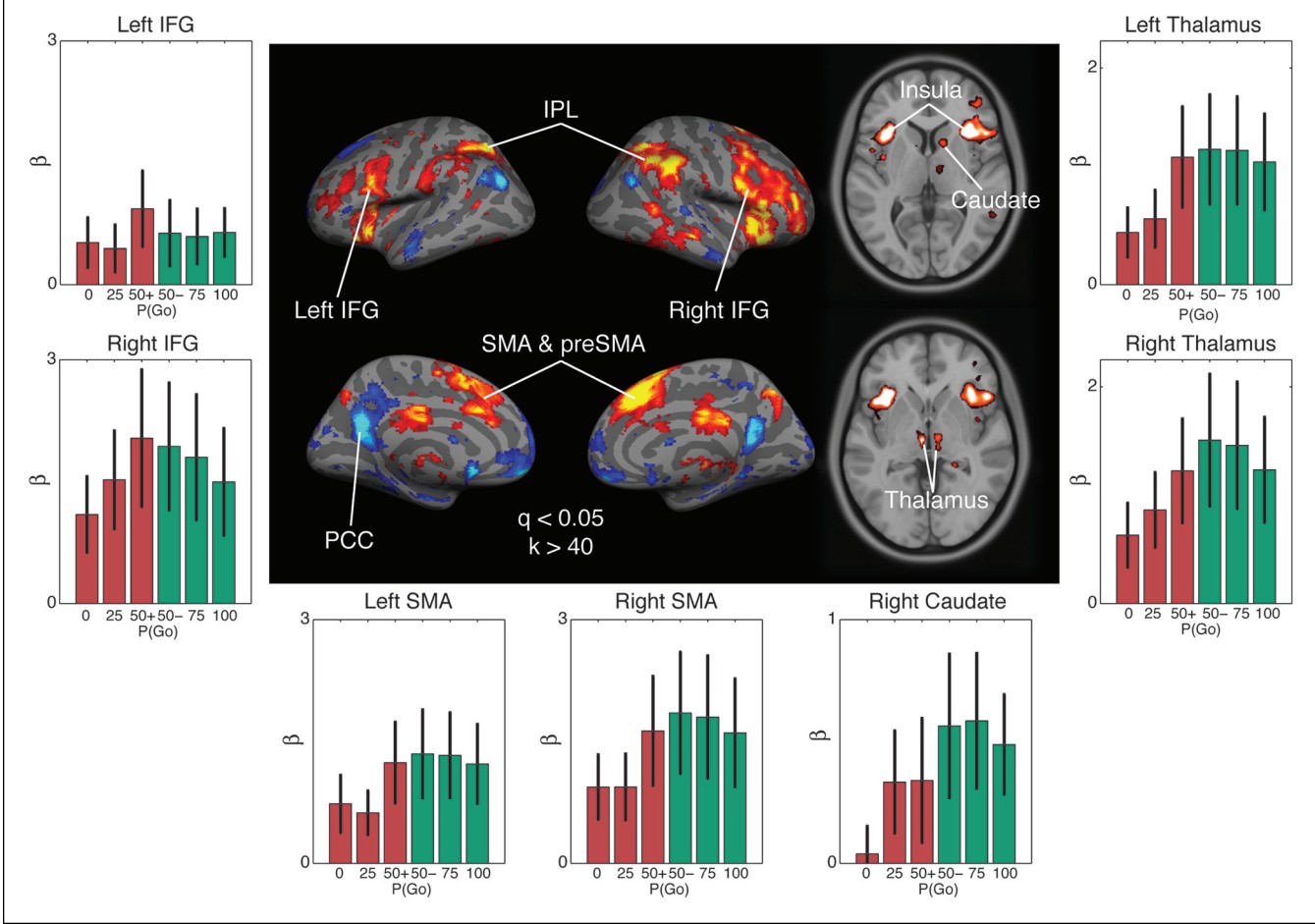

**Figure 8.** Observed interaction between trial outcome and response expectation on BOLD activation. Contrast maps for the comparison of no-go responses, modulated by go trial probability, against modulated go responses in the proactive task (center panel). Warm colors show areas where the modulation was more positive during no-go trials than go trials. This is the direction predicted by the drift modulation model (*Figure 7A*). All voxels are corrected to a false discovery rate of 0.05 (i.e., q <0.05). Region of interest (ROI) clusters were thresholded to a minimum of 40 continuously connected voxels (i.e., k >40). The side and bottom panels show individual responses for the go trial probability condition in seven ROIs. Red bars illustrate BOLD responses during trials where no key press was registered (no-go trials) and green bars show BOLD fits to trials where a response was registered (go). BOLD: blood oxygenation level dependent; AMA: supplementary motor area; preSMA: pre-supplementary motor area; IFG: inferior frontal gyrus; IPL: inferior parietal lobule; PCC: posterior cingulate cortex.

The following figure supplements are available for Figure 8:

**Figure supplement 1.** In-scanner EMG protocol.

threshold in the traditional drift-diffusion models without a dynamic pressure on the drift rate. Thus, activation patterns in distributed corticobasal ganglia circuits are largely consistent with the predicted responses from our computational model of competing decision dynamics on execution process drift rates.

## Discussion

The current results provide two novel insights into the dynamics of inhibitory control. First, a DPM, where the initial state of the action cancellation signal depends on the current state of the execution decision, provides a much better fit to reactive stopping performance than models where these two processes are independent. As we showed, this dependency is similar to the interactive race model, proposed by *Boucher et al. (2007)* to describe inhibitory control of saccadic eye movements, in

which an accumulating go signal is directly suppressed by a much stronger stop signal, thereby deterministically canceling subthreshold actions. In this model the critical parameter for determining stopping performance is the delay between a stop cue and when the stop signal is actually registered. In contrast, the critical parameters in our model are the race between the execution and braking processes and the state of the execution process when the stop signal occurs. This dependency is largely consistent with the circuit-level architecture of corticobasal ganglia pathways (*Figure 1A*; *Utter and Basso, 2008*) where the direct (facilitation), indirect (suppression), and hyperdirect (brake) pathways converge at the primary output nucleus of the basal ganglia (i.e., the GPi in primates). Here we show, for the first time, that this dependency better fits behavioral data in the reactive task than the interactive race model, suggesting that there may not be equivalent control dynamics between oculomotor and manual response decisions.

Schmidt et al. (2013) found that, in rodents, an external stop cue reliably produces a burst of activity in the STN, but only results in a successful stop when this signal reaches the substantia nigra pars reticulata (similar to the GPi in primates). Consistent with classic race models of action cancellation, early activity of movement encoding striatal neurons was sufficient to predict whether hyperdirect stop signaling would reach the substantia nigra pars reticulata in time to prevent a response, rising rapidly on fast go and failed stop trials and more gradually on slow go and successful stop trials. Parts of this action cancellation signal from the STN may also be relegated via arkypallidal pathways (orange pathways in *Figure 1A*) that relay inhibitory signals back up to the striatal neurons, offering a route for hyperdirect signals to terminate action plans within the striatum itself (*Dodson et al., 2015*). Both temporal competition between hyperdirect and descending striatal signals at the GPi and cancellation via arkypallidal pathways are largely consistent with the DPM presented here.

Our modeling and imaging results also provide novel insights into the nature of proactive No-go decisions. Our framing of the execution process as a dynamic competition between direct and indirect pathways is motivated by the architecture of corticobasal ganglia pathways (*Figure 1A*), as well as recent electrophysiological and modeling evidence showing that cortical inputs to the striatum dynamically control the balance of direct and indirect signals throughout the basal ganglia (*Wei et al., 2015*). There are many possible routes for a competition between direct–indirect pathways with the basal ganglia. First, given that STN and globus pallidus projections terminate on common outputs, it is possible that a simple summation of inhibitory and excitatory signals at the GPi itself results in a separate race for disinhibiting pallidal projections to the thalamus. It is also possible that branching collaterals from direct pathway neurons that terminate on indirect pathway neurons in the globus pallidus (*Wei et al., 2015*; *Wu et al., 2000*) would allow for crosstalk between the two major striatal efferent pathways. Finally, as mentioned previously, indirect pathway output, at the level of the STN, may be relayed back up to striatal direct pathway cells via arkypallidal connections (*Dodson et al., 2015*). Disambiguating which of these mechanisms may proactively modulate the accumulation of evidence toward an execution decision requires careful physiological analysis within each pathway itself and is left to future studies.

By reconceptualizing the dynamics of action selection as an ongoing competition between direct (facilitation) and indirect (suppression) pathways, our model may explain some of the discrepancies in previous studies of inhibitory control. For example, proactive inhibition has been associated with hemodynamic responses in the dorsal striatum (*Majid et al., 2013*), interpreted as an increased recruitment of the indirect pathway to suppress an action. Yet, electrophysiological recordings in rodents find concurrent activation of both the direct and indirect pathways during action initiation and cancellation (*Cui et al., 2013*). Both of these findings are consistent within the context of our model, in which the relative concurrent activation of direct and indirect pathways determines rate of integration toward the execution decision threshold.

Previous studies have largely found that proactive control is implemented as a change in the distance to threshold (*Cavanagh et al., 2011b*; *Forstmann et al., 2010*; *Frank et al., 2015*; *Jahfari et al., 2012*; *Wei et al., 2015*); whereas in our model, these dynamics are captured by the rate of the execution process itself. While drift rate and threshold modulations have qualitatively dissociable effects in a standard two-choice drift-diffusion model—increasing the decision threshold leads to a speed-accuracy trade-off, whereas increasing drift rate has a positive effect on both outcome measures—this distinction does not hold true for our DPM. Rather, slowing the rate of the execution process leads to greater stopping accuracy at the expense of response speed. This is

consistent with speed-accuracy trade-off produced by raising the response threshold in traditional drift-diffusion models. Also, *Ratcliff and Frank (2012)* found that synthetic behavioral data generated by a basal ganglia neural network were best described by a drift-diffusion model with a decision threshold that collapsed at a rate that depended on the conflict between choices. Here we included a similar mechanism, but as a dynamic gain signal applied to the execution process, and compared models in which response uncertainty modulated its drift-rate, onset time, or the combination of these parameters. Our findings suggest that in the presence of these temporal dynamics, contextual factors like control demands impose a stronger influence on the rate of the growing urgency to respond than on the onset of the decision process. This finding is consistent with the context-dependent rate decay in the decision threshold found by *Ratcliff and Frank (2012)*. However, by capturing these dynamics in the accumulating process itself, the DPM allows for incorporating the effects of context in reactive (i.e., penalty and reward structure) and proactive (i.e., probabilistic expectations) control in a single unifying parameter (i.e., execution drift rate), while at the same time distinguishing between action cancellation and no-go decision processes.

Compared with stationary diffusion models, which predict positively skewed RT distributions, models that assume a temporal decay in the threshold or a nonconstant rate of accumulation predict response times that are more normally distributed (*Hawkins et al., 2015*). Indeed, both tasks produced RT distributions that appeared more normal in shape and lacked a positive skew, adding justification to the assumption of a dynamic bias in the execution process. Compared with the reactive task (*Figure 4—figure supplement 1*), however, proactive RT distributions (*Figure 6—figure supplement 1*) were much narrower, spiking near the end of the trial with a slight negative skew. Correspondingly, best-fit parameter estimates show a muc stronger influence of the dynamic bias signal on 'go' decisions in the proactive than reactive task, suggesting that gain adaptation is differentially recruited during these two forms of inhibitory control. Future studies will be necessary to investigate the contextual constraints and neural plausibility of nonstationary decision mechanisms, particularly with respect to their implementation in corticobasal ganglia networks.

Other models of behavioral and electrophysiological data in countermanding tasks, such as the anti-saccade task, have suggested that proactive constraints shift the onset of the decision process. This is particularly true in cortical neurons that plan the action (*Pouget et al., 2011*; *Purcell et al., 2010*). However, here we show that the onset modulation model did not fit the behavior or imaging results as well as the drift modulation model. Our model was specifically designed to capture constraints in corticobasal ganglia systems, which are upstream of motor execution neurons, such as those recorded in previous countermanding tasks. Thus, it is possible that the dynamics described here are the mechanism for the delayed onset of activity in downstream cortical motor areas, that is, the onset of action selection neurons happens when the decision processes from the basal ganglia reach their boundary, which becomes delayed as the rate of the execution process is decelerated by the indirect pathway. It is also well established by now that subjects do not rely on a single mechanism for incorporating context into the decision process (*Heitz and Schall, 2012*; *Standage et al., 2011*). While the pattern of observed BOLD responses was generally consistent with the drift modulation predictions, it did share some characteristics with the predictions of the onset modulation model (i.e., increasing activity with go trial probability on 'no-go' trials and greater overall activation on 'go' trials), allowing for the possibility that some subjects may alternate between strategies of modulating the drift and onset across trials or use both simultaneously. Only careful analysis of coincident electrophysiological recordings in key basal ganglia and cortical regions will fully confirm the compatibility of the two models.

Taken together, our results provide crucial evidence that reactive stopping ability depends on the competitive dynamics of ongoing proactive decision processes, suggesting these two forms of control arise from convergent, but separable control signals in the basal ganglia. This novel reconceptualization of how proactive no-go decisions are distinguished from reactive stopping is consistent with the growing body of evidence to suggest that inhibitory control is a multifaceted process comprised of separable, but not independent, control mechanisms.

# Materials and methods

## Participants

Neurologically healthy adults (behavioral study: N=60, mean age=22 years, 28 male; imaging study: N=20, mean age=25 years, 9 male) were recruited from the local university population. All procedures were approved by the local institutional review board at Carnegie Mellon University.

## Tasks

The general trial procedure for go trials was the same for all experiments. On each trial, participants would see a bar fill upwards toward a white target line on the screen. The bar would intersect the target line 500 ms after trial onset. Participants could stop the bar at any point in the trial by pressing a button with their right hand (space key in behavioral study, index button on a response pad in the fMRI experiment) and were instructed to stop the bar as close to the target line as possible. The trial was terminated when the subject made a key press or when the bar passed the upper limit of the screen (reactive task, 650 ms; proactive task, 555 ms). On each trial a monetary bonus was given based on how close the top of the bar was to the target line. On baseline trials, this was determined as the inverse of the distance from the target. On caution trials, this was determined as the inverse of the squared distance. At the end of each trial, participants received visual feedback of their accuracy and monetary score.

In the reactive experiment, on a subset of trials (stop trials, n = 100) the bar would stop and turn red at one of five predefined points in its trajectory (the SSD). SSDs ranged from 200 to 400 ms (n=20 trials per SSD). On these trials participants were instructed not to produce a key press. Subjects were told that the color change was a cue that the bar had stopped. If subjects correctly withheld a response, then they would receive a bonus score. If subjects erroneously made a response, they received a penalty of equal value to the bonus during successful trials (100 points in baseline trials; 200 points in caution trials). The first block of trials (n=22) were all go trials so as to familiarize subjects with the timing of the response. On the remaining nine blocks (n=20 trials per block), the stop and go trials were randomly interleaved with a 50% probability. This ratio was selected so as to match the stopping probabilities with the proactive experiment.

In the proactive experiment, only one SSD was used (450 ms) for the entire experiment. On each trial the color of the bar indicated the probability that the bar would intersect with the target. Blue bars indicated 100% probability, red bars indicated 0% probability (i.e., always a stop trial), and shades of purple represented mixture probabilities. On each block (n=24 trials/block, 10 blocks total), each go trial probability was uniformly sampled four times. In order to detect late key presses on no-go trials, the program would continue monitoring for key presses for an extra 100 ms after the bar left the screen. The feedback structures for the baseline and caution conditions were the same as in the reactive experiment.

All experiments were run using Psychophysics Toolbox (version 3.0) in Matlab (R2012b) on a Linux platform (Ubuntu 12.04).

## Behavioral analysis

A repeated-measures ANOVA was performed to test for a statistically significant interaction between the effects of cued go trial probability (0%, 20%, 40%, 60%, 80%, 100%) and penalty structure (baseline, caution) in the mean probability of a no-go decision. The same test was performed for mean go RTs but collapsing across 0%, 20% and 40% go trial probability conditions to increase statistical power, as fewer responses were recorded in these conditions. To account for hardware limitations in precision of RT measurements, mean go RTs were calculated for all responses recorded prior to the earliest recorded 'no-go' outcome ( $<\sim$555 ms). Both dependent measures violated the sphericity assumption, thus all reported values were corrected accordingly using the Greenhouse-Geisser method. The PSE on the stop and no-go probability curves was estimated using logistic regression. Specifically the probability of stopping or not-going, $y$, was modeled as a function of cue conditions, $x$.

$$y = \lambda x \ \lambda_0 \qquad (1)$$

For the reactive task, *x* was the stop-signal delay. For the proactive task, *x* was the probability of a go trial. The PSE was then calculated using the constant, $\lambda_0$, and slope, $\lambda$, parameters from this regression model.

$$PSE = -1\frac{\lambda_0}{\lambda} \qquad (2)$$

## Drift diffusion process models

For all simulations, we designed an extension of the standard drift-diffusion model which assumes evidence for competing choices is stochastically sampled and accumulated until terminating at one of the choice thresholds, determining the decision outcome and response time. Our modified model assumes that competing facilitatory (i.e., direct) and suppressive (i.e., indirect) signals are integrated into a single execution process ($\theta_e$). The linear drift and diffusion ($\varphi_e$) of the execution process is shown by the stochastic differential equation in *Equation 3*, accumulating with a mean rate of $v_e$ (i.e., drift rate) and a standard deviation described by the Wiener diffusion process (e.g., white noise) with diffusion constant $\sigma$. The execution process is fully described by *Equation 4* in which the linear accumulation described by *Equation 3* is scaled by a dynamic bias signal $\gamma$, modeled as a hyperbolic function of time with gain *xb*.

$$d\varphi_e = v_e dt \ \sigma dW \qquad (3)$$

$$\theta_e(t) = \varphi_e(t) \cdot \gamma(t) \qquad (4)$$

The execution process begins accumulating after an onset delay of *tr*. It is worth noting that it is possible for the onset delay parameter to capture both pre- and postdecision delays (i.e., sensory encoding and motor execution delays).

A response and RT is recorded if $\theta_e$ reaches the execution boundary (*a*) before the end of the trial window (*b*) and before the braking process reaches the lower (0) boundary (see below). In the event of a stop cue, the braking process ($\theta_b$) is initiated at the current state of $\theta_e$ with a negative drift rate ($-v_b$). If $\theta_b$ reaches the 0 boundary before $\theta_e$ reaches the execution boundary no response or RT is recorded from the model.

For the reactive stopping task we fit three different models where the dynamics of $\theta_e$ and $\theta_b$.

- Dependent process model: Here the change in $\theta_b$ over time is given by *Equation 5*, the same stochastic differential equation that expresses the temporal dynamics of $\theta_e$ but with a negative drift rate ($-v_b$) and in the absence of the dynamic bias signal (*Equation 3*). The dependency between $\theta_b$ and $\theta_e$ in the DPM is described by the conditional statement that the initial state of $\theta_b$ (occurring at t = SSD) is equal to the state of $\theta_e$ at t = SSD

$$\begin{aligned} d\varphi_b &= v_b dt \ \sigma dW \\ \theta_b(SSD) &= \theta_e(SSD) \end{aligned} \qquad (5)$$

- Independent process model: Here the change in $\theta_b$ over time is given by *Equation 6*, the same equation as *Equation 5*, but with positive a drift rate ($v_b$). The independence between $\theta_b$ and $\theta_e$ in this model is described by the conditional statement that the initial state of $\theta_b$ (occurring at t = SSD) is equal to the initial state of $\theta_e$ (occurring at t = *tr*).

$$d\theta_b = v_b dt \ \sigma dW \qquad (6)$$

$$\theta_b(SSD) = \theta_e(tr) = 0 \qquad (7)$$

- Interactive race model: We modified a model originally proposed by *Boucher et al., 2007* in which a stochastically drifting process executes a response upon reaching a threshold. On stop cue trials, a response is suppressed by the onset of a braking process that directly inhibits the execution process, preventing it from reaching the action threshold. The original model was selectively parameterized to fit distinct physiological properties of the oculomotor system in order to test specific predictions about the neural dynamics of this network during an inhibitory saccade task. Here we adapt the model to preserve the basic mechanisms by which a response is executed and suppressed and is a useful model to compare against the DPM. The

interactive race model is composed of a single absorbing bound (*a*), marking the distance from 0 that the execution process must travel in order to elicit a response. The execution process follows *Equations 3* and *4* above. The braking process in this model is also parameterized by its onset delay (*sso*) and drift rate (*$v_b$*), so that the absolute onset of the braking process occurs at *tr* ms from the start of the trial and the absolute onset of the braking process occurs at *sso* ms from the presentation of the stop cue (*$ss_{tr}$* = *SSD sso*). Both execution and braking processes have positive drift rates. On trials in which the execution process has not reached the action threshold by *$ss_{tr}$*, the accumulating signal of the braking process is subtracted from the execution signal until the end of the trial.

We also compared different models of contextual modulation in which penalty and proactive cuing either modulated the drift rate, onset time, or boundary height of the execution process. All contextual modulation models in the reactive task were fit using the DPM framework (*Equations 3–5*) with two free parameters in the drift (*$v_b$, $v_c$*), onset (*$tr_b$, $tr_c$*), and boundary (*$a_b$, $a_c$*) modulation models and four free parameters in the combined drift and onset modulation model (*$v_b$, $v_c$, $tr_b$, $tr_c$*). The corresponding models fit to the proactive task contained six free parameters (one each go trial probability cue) in the drift (*$v_0$–$v_{100}$*), boundary (*$a_0$–$a_{100}$*), and onset (*$tr_0$–$tr_{100}$*) modulation models and 12 free parameters in the combined drift and onset model (*$v_0$–$v_{100}$; $tr_0$–$tr_{100}$*). We report two complexity-penalized goodness-of-fit statistics to account for differences in the number of parameters of alternative models, AIC and BIC.

## Model fitting

All models were fit to the behavioral data by minimizing a cost function equal to the sum of the squared and weighted errors between vectors of observed and simulated RT quantiles (0.1,0.3,0.5,0.7,0.9) and response probabilities, that is, stop probability curve in the reactive task and the no-go probability curve in the proactive task. Weights applied to the RT quantiles were calculated by estimating the standard error for each of the five quantiles (Maritz and Jarrett, 1978) and then dividing the interpolated median standard error by that of each quantile. For the response probabilities, weights were calculated by taking the standard deviation for each condition across subjects and then dividing the mean of those standard deviations by that of each condition. This approach represents the variability of each value in the vector as a ratio (*Ratcliff and Tuerlinckx, 2002*), where values closer to the mean are assigned a weight close to 1, and values associated with higher variability a weight <1, lower variability a weight >1.

The cost function for all models fit to the reactive task data is shown in *Equation 8* below. This included the probability of a response on go trials ($P_g$), the stop accuracy at each of the five SSD conditions ($P_{ssd}$), the five RT quantiles for correct responses ($Q^c$) and five RT quantiles for error responses ($Q^e$) for both the baseline and caution conditions (j) separately. Error RTs (i.e., responses on trials when the stop cue was presented) were collapsed across all SSDs due to there being few trials in which the subject made a response in the early SSD conditions. The weights of the correct ($w^c$) and error ($w^e$) RT quantiles were multiplied by the respective probability of a response on correct ($P_{corr}$) and error ($P_{err}$) trials. All variables with the hat accent are model-predicted values and are subtracted from the corresponding empirical subject-averaged statistic.

$$Cost_{Reactive} = \sum_{j=1}^{2} [w_g \cdot (P_g^j - \widehat{P}_g^j)^2 \sum_{d=1}^{5} w_d \cdot (P_d^j - \widehat{P}_d j)^2$$

$$P_{corr}^j \cdot \sum_{i=1}^{5} w_{ij}^c \cdot (Q_{ij}^c - \widehat{Q}_{ij}^c)^2 \; P_{err}^j \cdot \sum_{i=1}^{5} w_{ij}^e \cdot (Q_{ij}^e - \widehat{Q}_{ij}^e)^2]$$

(8)

Proactive models were fit using a similar cost function that included the probability of a response in each of the six probabilistic cue conditions ($P_{prob_i}$), five RT quantiles calculated from trials in which the cue indicated a go probability higher that 50% ($Q_{hi}$) and five RT quantiles calculated from trials in which the cue indicated a go probability of lower than 50% ($Q_{lo}$). The weights applied to high ($w_h$) and low ($w_l$) RT quantiles were multiplied by the respective probability of a response in high ($P_{high}$) and low ($P_{low}$) go probability conditions.

$$cost_{\text{Proactive}} = \sum_{j=1}^{6} w_p \left( P^j - \widehat{P}^j \right)^2 P_{high} \sum_{h=1}^{5} W_h \left( Q_h - \widehat{Q}_h \right)^2 P_{low} \sum_{i=1}^{5} w_i \left( Q_i - \widehat{Q}_i \right)^2 \qquad (9)$$

The response times were split up in this manner to account for the low trial counts in the lower go trial probability conditions. During the model fits, this was performed by allowing free parameters in the model to vary across all six cue conditions, in order to calculate the predicted response probabilities. All trials in which the execution process reached threshold before the time limit were then collapsed into two vectors, one containing simulations from the low go trial probability conditions and the other from high go trial probability conditions. RT quantiles were then calculated for each aggregate vector of RTs and appended to the vector of response probabilities before being submitted to the cost function. All variables with the hat accent are model-predicted values and are subtracted from the corresponding empirical subject-averaged statistic.

To ensure that parameter estimation did not terminate prematurely, models were fit to the data using a combination of global and local minima optimization. First, an initial set of parameters was obtained for each model type by leaving all parameters free and using the basin hopping algorithm provided by SciPy (*Blanco-Silva, 2013*) to identify a global error basin. The best fitting parameters are then optimized using a Nelder–Mead simplex algorithm (*Nelder and Mead, 1965*) which explores the basin reached in the previous step for its local minimum. Basin hopping seeks to identify a set of parameters at the global minimum of a cost function by stochastically searching the parameter space for error basins, which is especially desirable when there is a high degree of the variability in the model error. However, stochastic search methods are not well suited for fine-grained error minimization which is best achieved by gradient based methods like the simplex algorithm. For reactive models, this average cost function is similar to that shown in *Equation 8* but without the summation over j conditions and with all empirical statistics representing the average value collapsing baseline and caution conditions. For proactive models, this cost function becomes a single response probability, taken as the average response probability across cues, and five RT quantiles calculated by averaging the quantiles for the high and low go trial probability conditions. At this stage the basin hopping algorithm is iterated a maximum of 100 times or until it returns 40 failed attempts to find a new global minimum

Following parameter optimization to the average data, alternative models were tested by allowing only one or a few of these parameters to vary across conditions while holding all others constant. First, the average parameter estimates are submitted to another global optimization procedure in which all parameters are held constant except for the parameter of interest that is optimized individually to each condition of interest using the same basin hopping parameters as above. This is performed prior to the final simplex optimization to avoid initiating the parameter(s) at the same value for each condition being fit, as we found this to increase the chances that the Nelder–Mead simplex terminates very close to its initial state and overall generally worse fits. The final simplex minimization was run holding all parameters constant except for those of interest that were left free to vary across conditions. For all models, this procedure was run from start to finish at least three times and the model with the lowest AIC was selected. This includes all compared models of contextual modulation in the reactive and proactive tasks, meaning that constant parameters are held constant across conditions, but not models. By running every model through the same global and local optimization routine multiple times, this avoids the issue of biasing model selection to favor those that are simply better able to explain the data within the constraints of a single set of constant parameters. Thus, parameter estimates listed in *Tables 1 and 2* for parameters held constant across conditions reflect the best-fit parameters to the average data for the best-of-three model.

Example data as well as model code, simulations, cost function weights, and animations can be found here: https://www.github.com/coaxlab/radd

## Model-simulated BOLD activity

In addition to fitting different models of proactive control to behavioral data, we evaluated these models on their ability to make general predictions regarding the change in the BOLD response across probabilistic cues on go and no-go trials. Based on the assumption that the neural activity in regions involved in a competitive execution decision will reflect information about the time and distance of the accumulation process, we generated BOLD predictions from each of the models using

the following procedure. For each model, the best fitting parameter estimates from the proactive task were used to simulate 100 trials for each go trial probability condition. Because we were interested in comparing fundamental differences in how each model predicted BOLD activity across conditions rather than trial-wise or subject-wise predictions, simulations were performed using the best fit parameters from the behavioral experiment rather than performing separate fits to behavior from the scanning session. This was done to take advantage of the fact that the behavioral effects of the task were similar inside and outside the scanner, but with significantly more trials/observations in the behavioral-only session.

The simulated trials were then separated by trial outcome ('go' trials in which the execution process reached threshold and 'no-go' in which the execution process failed to reach threshold. To approximate the same experimental conditions tested in the imaging analysis, 'go' trials in the lower go trial probability conditions ($\leq$60%) were binned together and all 'no-go' trials in the higher go trial probability conditions ($\geq$40%) were binned together. For each condition the predicted BOLD magnitude was calculated by taking the cumulative sum of the full execution process over the length of each trial, then averaging across trials. By taking the cumulative sum of the execution process this effectively represents the distance-to-threshold and time-to-threshold of each trial as a single area-under-the-curve estimate. Taking the average area-under-the-curve across trials can then be compared with the observed magnitude of the BOLD response in the corresponding condition.

## fMRI acquisition and analysis

All imaging data were collected in a 3T Siemens Verio MRI at the Scientific Imaging and Brain Research Center on the Carnegie Mellon University campus. All functional images were collected with echo planar imaging sequence (TE = 20 ms; TR = 1500 ms). Thirty contiguous slices (3.2 mm x$\times$ 3.2 mm $\times$ 4 mm) were collected in an ascending and sequential fashion parallel to the anterior and posterior commissures. One subject was removed from analysis due to an error in saving the output files for the scan run. One subject failed to complete the reactive experiment due to time constraints. All experiments utilized a rapid event related design with a jittered intertrial interval sampled from an exponential distribution (range 4–18 s, mean = 8 s) with 334 volumes per run. In the proactive experiment, a total of four scan runs were collected with 60 trials collected for each go trial probability condition. All trials were preceded by a 300 ms onset cue

During scanning electrical activity of the FDI muscle of the right hand was recorded using an MR compatible EMG sensor (400 Hz sampling rate). Sensors were put over the lateral and medial portions of the FDI muscle, with the ground electrode placed on the bone of the wrist (*Figure 8—figure supplement 1*). The onset and offset of recording were time-locked to the onset of the functional scans. In a subset of subjects (N = 6) the electrode placement slipped or a recording buffer error truncated data collection. Thus EMG data for these subjects was not included in the final analysis. The muscle activity for all trials was smoothed using a Gaussian smoothing kernel (4 s FWHM) and time-locked to the onset of each trial. Significant muscle activity was determined within each trial when the root mean squared EMG signal exceeded 3 standard deviations of the pre-trial signal.

All fMRI analysis was performed with SPM8 (http://www.fil.ion.ucl.ac.uk/spm/software/spm8/). Prior to analysis, the echo planar images for each participant were realigned to the first image in the series and corrected for differences in the slice acquisition time. All images were then coregistered to MNI-space (Montreal Neurological Institute) using a nonlinear spatial normalization approach (Internation Consortium for Brain Mapping-152 space template regularization, 16 nonlinear iterations) and smoothed using a 4 mm isotropic Gaussian kernel. Estimates of task-related responses at each voxel were determined using a reweighted least squares GLM approach (*Diedrichsen and Shadmehr, 2005*) that minimizes the influence of movement related noise in the signal. Statistically significant effects were determined using a false discovery rate threshold across all voxels of 0.05 (q < 0.05).

Clusters of 40 or more continuously active voxels were extracted from the parametric ROIs and their condition-wise BOLD responses are reported in *Supplementary file 1*. Seven areas were identified in the whole brain parametric GLM analysis as being of particular interest and selected for follow-up ROI analysis. To avoid circular inference, we used the Harvard–Oxford cortical and subcortical atlases (*Desikan et al., 2006*; Frazier et al., 2005) to anatomically identify the right caudate nucleus and bilateral inferior frontal gyrus (operculum), SMA, and thalamus. Using a fourfold cross validation, voxels with a significant (p <0.005, uncorrected) positive no-go–go parametric

modulation were identified within each anatomical ROI using three-fourths of the sample. Then the regression coefficients for each condition were extracted from this subset of voxels within the ROI in the remaining subjects.

## Acknowledgements

The authors would like to thank Ian Greenhouse and Lori Holt for their helpful comments on early drafts of this manuscript. This research was funded by the Pennsylvania State Department of Health Formula Award SAP4100062201 and by the Army Research Laboratory under Cooperative Agreement Number W911NF-10-2-002. The views and conclusions contained in this document are those of the authors and should not be interpreted as representing the official policies, either expressed or implied, of the Army Research Laboratory or the US Government. The US Government is authorized to reproduce and distribute reprints for Government purposes notwithstanding any copyright notation herein.

## Additional information

### Funding

| Funder | Grant reference number | Author |
| --- | --- | --- |
| Pennsylvania Department of Health | Formula Award SAP4100062201 | Timothy Verstynen |
| U.S. Army Research Laboratory | Cooperative Agreement Number W911NF-10-2-002 | Timothy Verstynen |

The funders had no role in study design, data collection and interpretation, or the decision to submit the work for publication. The views and conclusions contained in this document are those of the authors and should not be interpreted as representing the official policies, either expressed or implied, of the Army Research Laboratory or the U.S. Government. The U.S. Government is authorized to reproduce and distribute reprints for Government purposes notwithstanding any copyright notation herein.

### Author contributions

KD, TV, Conception and design, Analysis and interpretation of data, Drafting or revising the article; BL, Acquisition of data, Drafting or revising the article; TM, Acquisition of data, Analysis and interpretation of data

### Ethics

Human subjects: Neurologically healthy adults were recruited from the local university population. All procedures were approved by the local institutional review board at Carnegie Mellon University. All research participants provided informed consent to participate in the study and consent to publish any research findings based on their provided data.

## Additional files

### Supplementary files

• Supplementary file 1. Table of significant clusters for the no-go parametric minus go parametric contrast shown in *Figure 8A*. Coordinates are centers of mass for the cluster in MNI-space. N is the number of voxels in each cluster. Values in the left six columns show average condition-wise (general linear model) GLM coefficients and standard deviation across subjects is in parentheses.

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
