## [Decision Letter]

Thank you for submitting your work entitled "Competing basal-ganglia pathways determine the difference between stopping and deciding not to go" for peer review at *eLife*. Your submission has been favorably evaluated by Timothy Behrens (Senior editor), Michael Frank (Reviewing editor), and three reviewers.

The reviewers have discussed the reviews with one another and the Reviewing editor has drafted this decision to help you prepare a revised submission.

Summary:

The authors present a novel sequential sampling model of action cancellation in a stop-signal task, whereby one decision variable initially drifts toward a threshold for action, and a second decision variable, initiated at the stop-signal delay then competes with the first to reach an alternate, braking threshold. They show that this model captures behavior in a reactive stop-signal task better than alternatives in which the stopping process starting point is independent of the initial decision variable. Finally, they show that neural activity in plausible convergence/output structures for the decision process (caudate, thalamus) is consistent with patterns predicted by their model.

Essential revisions:

This paper does a nice job of bringing together a number of computational and methodological approaches in an effort to better characterize how cognitive control unfolds in a stop signal-like task. The findings themselves have the potential to build on earlier work in important ways, particularly the modeling insights regarding the dependence between execution and braking; the comparisons between onset and drift-based models; and the neural patterns in caudate and thalamus. However, each of these is difficult to interpret properly without comparison to alternative models, which will also require fitting RT distributions rather than individual point estimates. Additional details related to the choice of model/parameters, and imaging assumptions are also needed. Thus a revision will require a good deal of work but we are hopeful that this can be accomplished.

1) Please compare the model to existing alternative models of response inhibition such as Boucher et al. which bears some resemblance to the current model and hence should be acknowledged, and the similarities and differences need to be made more explicit. In particular, the following model comparisons would be greatly useful, particularly "A"). While it may be out of scope to rigorously test each of the other alternatives, where possible it would be great if they could at least comment on their viability as an alternative (for example some may be able to be ruled out based on qualitative features of the data without necessarily fitting each model to the entire dataset). This would provide better insight into how wide or narrow the space of models is that could provide a similarly good account of the current data.

A) An approximation to the Boucher model would be to implement a version of your model in which execution and braking combine into a single decision variable, rather than separate DV's in parallel DDMs (this single-DV model would seem to be even closer to the authors' proposal of integration onto a single pool of neurons, particularly given the presumption that the execution drift is itself a combination of two pathway inputs);

B) A model in which the decision threshold for the initial process is altered by the stop-signal, as suggested by Wiecki & Frank's (2013) model in which the Go-NoGo pathways contribute positive and negative evidence for and against alternative responses but where the decision threshold on the Go-NoGo difference is modulated by STN activity given conflicting sources of evidence. This was applied to model various selective response inhibition tasks as well as stop signal tasks, including the cited Schmidt et al. data. (See also Ratcliff & Frank (2012) for DDM simulations linking STN to threshold changes, and various empirical studies showing cortical- >STN involvement in decision threshold regulation in conflict tasks (e.g. Cavanagh et al., 2011, Green et al. 2013)). This is also related to the more general work in psychology and neuroscience suggesting that probabilities affect baseline-threshold distance (e.g. see Carpenter & Williams, 1995; Platt & Glimcher 1999). Thus you could add a model in which the "colour of the bar" affects the threshold as an alternative model, and consider the relation to the above findings/models in Discussion.

C) A model in which drift remains constant past the SSD but braking is represented by a sudden change in threshold;

D) A model in which braking drift rate (v_s) is conditional on SSD;

E) A model in which both drift and onset are influenced by proactive parameters (mentioned in the Discussion);

F) A single-boundary model. The execution threshold has a straightforward interpretation when considering implementations of the DDM and related models in other domains, but the braking threshold (i.e., commitment to non-action) is less straightforward to interpret. While this has precedent in earlier race models in the action cancellation literature, I think it would be fair for someone to wonder whether it would be more parsimonious to implement the current model with a single absorbing boundary (execution) rather than two? Could the authors comment on the potential value in having this lower boundary as a termination point (rather than, e.g., simply allowing paths to cross it and return, and/or simply imposing this as a minimum on theta values)?

2) All of the reviewers were unsatisfied with the fitting methods which collapsed the RT to a single point estimate (the mean). This collapses across important information in the RT distribution, and the history in modeling choice RT with the full distributions has shown that the shape of these distributions is particularly important for making inferences about underlying cognitive processes (Ratcliff has many excellent papers on this). We would be more comfortable interpreting the conclusions made (e.g. that striatum affects drift rate rather than thresholds) with more serious consideration given to the fitting. And this will likely be especially important for disentangling the various alternatives noted above in point 1. Although you might not have the full likelihood of your model that would easily allow full Bayesian parameter estimation as in the HDDM, there are alternative approaches, the most straightforward of which would be to jointly optimize the quantiles and response proportions (i.e. find parameters that minimize a cost function equal to the summed (and appropriately weighted) differences between the two sets of quantiles, and response probabilities). Another approach would be to construct the likelihood of your model using simulation methods, see a recent PBR paper (by Turner, with code available). Aside from disentangling alternative models with more sensitive RT distributions, there really isn't need for a stochastic model if you only fit the mean, and the bootstrapping process doesn't help here (just effectively puts a kernel over the summary statistics with variance proportional to the number of points you resample). Also please include tables of average fit parameters across the different studies/conditions. Finally when performing model comparisons, why not use model fit statistics instead of paired t-tests?

3) Please clarify the how and why the computational models of the pro-active task make particular predictions for the BOLD signal. In the subsection “Model-Simulated BOLD Activity”: We did not fully understand how the predicted BOLD was obtained. Were the predictions not made for individual trials, but for a type of trials (e.g. with particular bar colour)? If so, we suppose the prediction was made on the basis of simulating the model multiple times and taking average. Please clarify and provide all details of this procedure, e.g. how many times the simulation was repeated, what parameter values were used, etc. In Figure 6, please explain what the dashed lines denote, and how exactly the colour curves were obtained. Finally, could you please provide an intuition for why the models make these particular predictions?

4) The fMRI results seem to depend non-trivially on assumptions about BOLD signal decay. Can the authors provide clearer justification for the need for a decay term and the specific form it took for the different conditions? This currently comes off a bit ad hoc.

5) As described, the ROI analyses in the fifth paragraph of subsection “Accumulating Basal Ganglia Output During No-Go Decisions “raise concerns about circularity (i.e., the ROIs were essentially selected based on the interactions now being tested: differences in probability modulation by condition). The authors should either provide additional information about these analyses that allays these concerns or use appropriate measures (e.g., leave-one-subject-out extraction) to avoid potential circularities.

6) Some important aspects of the experimental details could be referred to earlier. For example, reward structure of successful withholding of response seemed important but was only found in the later methods due to the format.

---

## [Author Response]

*This paper does a nice job of bringing together a number of computational and methodological approaches in an effort to better characterize how cognitive control unfolds in a stop signal-like task. The findings themselves have the potential to build on earlier work in important ways, particularly the modeling insights regarding the dependence between execution and braking; the comparisons between onset and drift-based models; and the neural patterns in caudate and thalamus. However, each of these is difficult to interpret properly without comparison to alternative models, which will also require fitting RT distributions rather than individual point estimates. Additional details related to the choice of model/parameters, and imaging assumptions are also needed. Thus a revision will require a good deal of work but we are hopeful that this can be accomplished. 1) Please compare the model to existing alternative models of response inhibition such as Boucher et al. which bears some resemblance to the current model and hence should be acknowledged, and the similarities and differences need to be made more explicit. In particular, the following model comparisons would be greatly useful, particularly "A"). While it may be out of scope to rigorously test each of the other alternatives, where possible it would be great if they could at least comment on their viability as an alternative (for example some may be able to be ruled out based on qualitative features of the data without necessarily fitting each model to the entire dataset). This would provide better insight into how wide or narrow the space of models is that could provide a similarly good account of the current data.*

We have made the following structural change to the general design of the model that we report here:

In order to achieve a similar computation as the collapsing bound model (Ratcliff & Frank, 2012) and match dynamic properties of inhibitory control neurons in motor areas (Murakami et al. 2014), we now implement a dynamic parameter to the execution process that accelerates exponentially to the decision boundary. Comparing our new model fits, with this parameter, to the old model without this parameter, we found that the exponential property dramatically improved overall fits to the behavioral data. Rather than add complexity to the number of models tested, we kept this hyperbolic parameter constant across all models.

We also expand the range of alternative models compared in the reactive task. We now compare our dependent process model to two alternatives:

Traditional independent race model: This DDM follows the classic, independent two-process race model.

Interactive race model: As suggested by the reviewers, we now include a comparison to the interactive race model proposed by Boucher and colleagues (2007). This affords a comparison to another model that allows for an interaction between execution and braking processes.

Finally we now test several different models of the modulatory influence of context (e.g., reward feedback and certainty of a Go trial). The models we now explicitly test in the paper are:

• Drift modulation model: Where context impacts the drift rate of the execution process.

• Onset modulation model: Where context delays the onset of the execution process.

• Boundary modulation model: Where context elevates the height of the decision boundary.

• Drift + Onset: A combination of drift and onset modulation.

We elaborate these changes in more detail below and in the main text. However, the key finding that the drift modulation model with dependent processes still stands out as the best fit to the behavioral data and matches dynamics of the BOLD signal during proactive control.

*A) An approximation to the Boucher model would be to implement a version of your model in which execution and braking combine into a single decision variable, rather than separate DV's in parallel DDMs (this single-DV model would seem to be even closer to the authors' proposal of integration onto a single pool of neurons, particularly given the presumption that the execution drift is itself a combination of two pathway inputs);*

This is an excellent point. The model presented by Boucher and colleagues does in fact provide an alternative dependent process model with similar assumptions as the model we present here. We now implement a model with similar properties as the interactive race model in our diffusion process framework (see Experimental Procedures). Based on standard model fit metrics (i.e., AIC & BIC) our dependent process model still provides better fits than the interactive race model, mainly due to better fits on the RT distributions. Qualitatively, however, both models captured the behavioral patterns quite well. Thus, these results should not necessarily be taken as evidence against the interactive race model since Boucher and colleagues developed their model to describe inhibitory control dynamics in oculomotor movements. Given substantial differences in the underlying neural substrates between ocular and manual movement systems, it seems reasonable that eye movements can be controlled using different mechanisms than manual actions.

*B) A model in which the decision threshold for the initial process is altered by the stop-signal, as suggested by Wiecki & Frank's (2013) model in which the Go-NoGo pathways contribute positive and negative evidence for and against alternative responses but where the decision threshold on the Go-NoGo difference is modulated by STN activity given conflicting sources of evidence. This was applied to model various selective response inhibition tasks as well as stop signal tasks, including the cited Schmidt et al. data. (See also Ratcliff & Frank (2012) for DDM simulations linking STN to threshold changes, and various empirical studies showing cortical- >STN involvement in decision threshold regulation in conflict tasks (e.g. Cavanagh et al., 2011, Green et al. 2013)). This is also related to the more general work in psychology and neuroscience suggesting that probabilities affect baseline-threshold distance (e.g. see Carpenter & Williams, 1995; Platt & Glimcher 1999). Thus you could add a model in which the "colour of the bar" affects the threshold as an alternative model, and consider the relation to the above findings/models in Discussion.*

We had initially decided to ignore boundary threshold modulations because we had assumed that the effects of altering the boundary would be similar to altering the drift (i.e., increasing the duration of the decision process) and conceptually could be considered to capture the same variance. However, upon review of this model we do, in fact, see that the boundary modulation and drift-rate modulation models provide different predictions to both behavioral and BOLD data. We now include leaving the boundary threshold free as an alternative to the onset and drift modulation models of proactive control (e.g., Carpenter & Williams, 1995; Platt & Glimcher 1999). While the boundary modulation model did capture the behavioral data quite well, it still under-performed the drift modulation model in fits to behavioral data and made predictions that ran counter to the task-related fMRI dynamics observed in output pathways of the basal ganglia (i.e., thalamus). We should point out that the inclusion of the exponential parameter in the execution process drift rate implements a similar process as a collapsing bound on the decision times (Ratcliff & Frank, 2012). Thus, this parameter now allows for a computationally similar effect as dynamic modulation on the decision threshold. We now elaborate how the drift rate and the dependent process model share similarities to the dynamic boundary models in the Discussion.

*C) A model in which drift remains constant past the SSD but braking is represented by a sudden change in threshold;*

We considered this model, however, adding in a temporally-dependent shift in threshold increases the number of parameters in already complex system. It is also very similar in structure to the interactive race model that we now include in the comparisons. While we do agree that a nuanced comparison of models where the properties of the diffusion process are non-stationary within the window of an individual trial, this is beyond the scope of the questions asked here and would detract from the main point of the paper (i.e., distinguishing dynamics of proactive and reactive control). Therefore, we decided to not include this class of non-stationary models in the comparisons here.

*D) A model in which braking drift rate (v_s) is conditional on SSD;*

As with our point in reply to 1. C above, we also chose not to include this model for two reasons. First, this is also a non-stationary model. Second, there is physiological evidence to suggest that the timing of the drift rate is constant (Schmidt et al. 2013) or, at a minimum, not dynamic enough to modulate with respect to the timing of the SSD within an individual trial.

*E) A model in which both drift and onset are influenced by proactive parameters (mentioned in the Discussion);*

This is a good point. In fact, in the Discussion of the previous submission, we elaborated on a similar possibility where the drift modulation determines an onset shift of a separate decision process in cortex. We now include the combined modulation model. However, after adjusting for the number of varied parameters, we do not see a difference in the goodness of fit relative to the simple drift-rate bias model. We believe this is due to the presence of the exponential modulation of the drift rate that we now include in our model.

*F) A single-boundary model. The execution threshold has a straightforward interpretation when considering implementations of the DDM and related models in other domains, but the braking threshold (i.e., commitment to non-action) is less straightforward to interpret. While this has precedent in earlier race models in the action cancellation literature, I think it would be fair for someone to wonder whether it would be more parsimonious to implement the current model with a single absorbing boundary (execution) rather than two? Could the authors comment on the potential value in having this lower boundary as a termination point (rather than, e.g., simply allowing paths to cross it and return, and/or simply imposing this as a minimum on theta values)?*

This is indeed an interesting idea. However, we should point out that having a lower boundary for a process to race to is a necessary condition for a dependent process model to work. If we understand the logic of the single absorbing boundary model correctly, then it would be difficult to test the dependency between the execution and braking processes if there wasn’t a determined distance for the braking process to “win” (e.g., from the state of the execution process at SSD to the lower boundary). Also, there is now very compelling physiological evidence that the hyper-direct braking signal has an absolute threshold that can define the temporal and spatial (Schmidt et al. 2013) criterion that must be met by the braking process in order to cancel an action. However, we attempted to make the model more parsimonious by eliminating the starting-point parameter of the execution process.

*2) All of the reviewers were unsatisfied with the fitting methods which collapsed the RT to a single point estimate (the mean). This collapses across important information in the RT distribution, and the history in modeling choice RT with the full distributions has shown that the shape of these distributions is particularly important for making inferences about underlying cognitive processes (Ratcliff has many excellent papers on this). We would be more comfortable interpreting the conclusions made (e.g. that striatum affects drift rate rather than thresholds) with more serious consideration given to the fitting. And this will likely be especially important for disentangling the various alternatives noted above in point 1. Although you might not have the full likelihood of your model that would easily allow full Bayesian parameter estimation as in the HDDM, there are alternative approaches, the most straightforward of which would be to jointly optimize the quantiles and response proportions (i.e. find parameters that minimize a cost function equal to the summed (and appropriately weighted) differences between the two sets of quantiles, and response probabilities). Another approach would be to construct the likelihood of your model using simulation methods, see a recent PBR paper (by Turner, with code available). Aside from disentangling alternative models with more sensitive RT distributions, there really isn't need for a stochastic model if you only fit the mean, and the bootstrapping process doesn't help here (just effectively puts a kernel over the summary statistics with variance proportional to the number of points you resample). Also please include tables of average fit parameters across the different studies/conditions. Finally when performing model comparisons, why not use model fit statistics instead of paired t-tests?*

This is an excellent point and one that we debated when working on the original version of the manuscript. We now implement a more sophisticated fitting routine with the following properties:

• Use of the full RT quantiles in the fits, rather than single point estimates;

• A parameter weighting scheme on RT and accuracy features adapted from Ratcliff & Tuerlinckx, 2002;

• A revised parameter identification routine using a combination of global and local optimization procedures, including serial basinhopping and Nelder-Mead simplex gradient descent approaches.

This new method provides substantially better fits than the previous fitting routine. We also now use information criteria (i.e., AIC & BIC) to evaluate model fits, rather than t-tests on the chi-square distributions.

*3) Please clarify the how and why the computational models of the pro-active task make particular predictions for the BOLD signal. In the subsection “Model-Simulated BOLD Activity”: We did not fully understand how the predicted BOLD was obtained. Were the predictions not made for individual trials, but for a type of trials (e.g. with particular bar colour)? If so, we suppose the prediction was made on the basis of simulating the model multiple times and taking average. Please clarify and provide all details of this procedure, e.g. how many times the simulation was repeated, what parameter values were used, etc. In Figure 6, please explain what the dashed lines denote, and how exactly the colour curves were obtained. Finally, could you please provide an intuition for why the models make these particular predictions?*

We have significantly modified the BOLD simulations, description in the Experimental Procedures, and corresponding figure (now Figure 7). This includes a more elaborate discussion of how each pattern of mean responses occurs in the Results.

*4) The fMRI results seem to depend non-trivially on assumptions about BOLD signal decay. Can the authors provide clearer justification for the need for a decay term and the specific form it took for the different conditions? This currently comes off a bit ad hoc.*

This concern arises from the way we presented the BOLD simulations so as to seem more biologically plausible. In order to estimate the predicted BOLD response with each condition, we simulated the summed model activity both with and without a decay phase in order to ensure that this did not unjustly influence the comparison between model predictions. For all BOLD simulations in the initial submission the decay rate was set to match the simulated rate of rise for each condition individually. Thus, the addition of the decay rate led to an overall increase in the predicted magnitude of the BOLD response but did not influence the relative magnitudes between models or between conditions. That said, since including the decay term did not contribute meaningfully to the predictions of the model and we have eliminated this from the current BOLD simulations for clarity. We elaborate on this in both the main text and the figure caption for the new Figure 7.

*5) As described, the ROI analyses in the fifth paragraph of subsection “Accumulating Basal Ganglia Output During No-Go Decisions “raise concerns about circularity (i.e., the ROIs were essentially selected based on the interactions now being tested: differences in probability modulation by condition). The authors should either provide additional information about these analyses that allays these concerns or use appropriate measures (e.g., leave-one-subject-out extraction) to avoid potential circularities.*

We had originally assumed that circular inference was not an issue with the ROI extraction because we were using one GLM, i.e., parametric model, to identify voxels with which to extract parameters from a separate GLM, i.e., simple condition-wise model. But we do see how circularity could be inferred given that these two models are fit to the same data. Therefore, to be as conservative as possible we have modified our ROI extraction routine using a 5 fold cross-validation to identify significant voxels within anatomically defined ROIs and extract out their condition-wise BOLD responses. The overall results are generally the same with two major exceptions: 1) the significance of the modulation during the Go decision trials is trending in the direction predicted by the drift-rate modulation model, but is not statistically significant. However, this direction is still supportive of the drift-rate model. 2) The patterns of the right caudate response are no longer statistically significant. The fact that the patterns in the Thalamus remain largely unchanged, however, holds with our prediction that these effects should be observed in the output areas of the basal ganglia.

*6) Some important aspects of the experimental details could be referred to earlier. For example, reward structure of successful withholding of response seemed important but was only found in the later methods due to the format.*

We now include more general details about the design in both the behavioral and imaging experiments throughout the Results section so as to minimize the reader having to go back and forth between the Experimental Procedures and Results.